# Induction of inverted morphology in brain organoids by vertical-mixing bioreactors

Dang Ngoc Anh Suong[1,2], Keiko Imamura[1,2,3], Ikuyo Inoue[1,3], Ryotaro Kabai[4], Satoko Sakamoto[4], Tatsuya Okumura[4], Yoshikazu Kato [5], Takayuki Kondo[1,2,3], Yuichiro Yada[1,2], William L. Klein [6], Akira Watanabe[4] & Haruhisa Inoue [1,2,3,7✉]

Organoid technology provides an opportunity to generate brain-like structures by recapitulating developmental steps in the manner of self-organization. Here we examined the vertical-mixing effect on brain organoid structures using bioreactors and established inverted brain organoids. The organoids generated by vertical mixing showed neurons that migrated from the outer periphery to the inner core of organoids, in contrast to orbital mixing. Computational analysis of flow dynamics clarified that, by comparison with orbital mixing, vertical mixing maintained the high turbulent energy around organoids, and continuously kept inter-organoid distances by dispersing and adding uniform rheological force on organoids. To uncover the mechanisms of the inverted structure, we investigated the direction of primary cilia, a cellular mechanosensor. Primary cilia of neural progenitors by vertical mixing were aligned in a multidirectional manner, and those by orbital mixing in a bidirectional manner. Single-cell RNA sequencing revealed that neurons of inverted brain organoids presented a GABAergic character of the ventral forebrain. These results suggest that controlling fluid dynamics by biomechanical engineering can direct stem cell differentiation of brain organoids, and that inverted brain organoids will be applicable for studying human brain development and disorders in the future.

[1] Center for iPS Cell Research and Application (CiRA), Kyoto University, Kyoto, Japan. [2] iPSC-based Drug Discovery and Development Team, RIKEN BioResource Research Center (BRC), Kyoto, Japan. [3] Medical-risk Avoidance based on iPS Cells Team, RIKEN Center for Advanced Intelligence Project (AIP), Kyoto, Japan. [4] Graduate School of Medicine, Kyoto University, Kyoto, Japan. [5] Mixing Technology Laboratory, SATAKE Chemical Equipment Manufacturing Ltd., Saitama, Japan. [6] Department of Neurobiology, Northwestern University, Evanston, IL 60208, USA. [7] Institute for Advancement of Clinical and Translational Science (iACT), Kyoto University Hospital, Kyoto, Japan. ✉email: haruhisa@cira.kyoto-u.ac.jp

Clarifying the principles regarding how brain cells arise and assemble tissue is important for understanding brain development and disease mechanisms. Model animals and genomic studies help us to analyze brain structure and evolution. However, there has been no method to study the hominid brain directly due to a lack of appropriate approaches. Brain organoid technology using human induced pluripotent stem cells (iPSCs) enables us to use a brain-like structure for research, and provides many insights into the development and diseases of human brain[1,2]. Brain organoids from human iPSCs retain these outstanding advantages in preparing various portions of brain and discrete cell types with neuronal circuitry[3–10], and a variety of brain organoid protocols have been developed. However, contributing factors to self-organization-driven formation of brain organoids are still not fully understood, and little is known about the effect of mechanical forces[11–15] except by orbital mixing on the formation of brain organoids from human iPSCs.

Organoids are cultured as three-dimensional cell aggregates in floating condition using shakers or bioreactors[1,15], and various mechanical stimuli including shear stress and turbulent energy, as well as energy dissipation affect cell differentiation and organoid formation. There have been some studies regarding the effects of mechanical forces on organoid development[16–18]. In the present study, we analyzed the effect of mechanical forces by vertical mixing on the formation of brain organoids. We used a vertical mixing bioreactor, which mixes culture medium with minimized shear stress, and maintains a uniform and stable environment by continuous monitoring of culture conditions, including stable temperature, pH, and dissolved oxygen concentration. As a consequence, the brain organoids generated by vertical mixing showed an inverted structure with neurons localizing at the center of organoids and covering outside by neural progenitors. This result proved that mechanical forces by biomechanical engineering contributed to human iPSC-derived brain organoid structures with enlarged and homogeneous areas of neurons inside inverted brain organoids, applicable for studying human brain development and disorders.

## Results

### Generation of inverted brain organoids by vertical mixing.
We generated brain organoids using orbital shakers for orbital mixing or reciprocal vertical bioreactors for vertical mixing (Fig. 1a, Supplementary Fig. 1). The structure of organoids after two months of culture by vertical mixing showed a different structure compared with that of organoids by orbital mixing. By orbital mixing, many neural tube-like structures were generated with SOX2-positive cells surrounded by MAP2-positive neurons. These structures were randomly distributed in brain organoids, showing an inside-out order with the neuron layer in peripheral regions and the neural progenitor layer in non-peripheral regions of organoids (Fig. 1b). In contrast, brain organoids generated by vertical mixing showed inverted structure, with the neural progenitor layer in the peripheral region and the neuronal layer in the center region of organoids (Fig. 1c). These inverted brain organoids contained a uniform area consisting of neurons in the middle of the structure. In vertical-mixing organoids, the area of SOX2-positive cells in the peripheral region, which is defined as within 100 μm from the edge of brain organoids, was significantly larger compared to that in orbital-mixing organoids (Fig. 1d).

Next, we investigated the layer formation of the cerebral cortex by analyzing cortical neuron markers. Inverted layer formation was observed in vertical-mixing organoids in comparison to those in orbital-mixing organoids (Fig. 1e, f). Alteration of the apical-basal order of layer formation was revealed by immunostaining with anti-N-CADHERIN, a marker for apical membrane, an

antibody (Fig. 1g, h). We investigated the effect of Matrigel for organoid structure by removing the Matrigel embed step for the orbital organoids, and found that the removal of Matrigel did not promote the generation of inverted structure (Supplementary Fig. 2a). Therefore, we consider that vertical mixing mainly contributes to the production of inverted brain structure, although removal of Matrigel may contribute to the inversion. These findings suggested that brain organoids by vertical mixing exhibited a special structure with an inverted inside-out pattern of layer formation, referred to as inverted brain organoids (Fig. 1i). We also found that inverted organoids were generated when orbital mixing was initiated on Day 15 following vertical mixing (Supplementary Fig. 2b), suggesting that the initiation of vertical mixing at an early stage of differentiation would be crucial for the formation of inverted organoids.

### Functional analysis of inverted brain organoids.
To investigate the functional maturation of inverted brain organoids, we evaluated neuronal activity by electrophysiological analysis. Inverted brain organoids dissociated to clumps on Day 56 were cultured on MEA chips for 6 additional weeks (Fig. 2a), and their spontaneous extracellular field activity was recorded. A representative phase-contrast image of a sample is presented (Fig. 2b). Spontaneous firing and synchronized burst firing were detected by recording, which indicated that the organoids harbored neuronal networks. To evaluate their synaptic functions, the response to compounds including GABA receptor antagonist and glutamate receptor antagonists were investigated. GABA receptor antagonist, pircrotoxin (PTX), increased the spike frequency and array-wide spike detection rate (ASDR), and a selective N-methyl-D-aspartate (NMDA) receptor antagonist, D-2-amino-5-phosphonopentanoate (AP-5), and AMPA/kainate glutamate receptor antagonist, 6-Cyano-7-nitroquinoxaline-2,3-dione (CNQX), inhibited the firing (Fig. 2c, d). These data suggest that inverted brain organoids exhibit functional maturity.

### Computational simulation of fluid dynamics.
To understand the mechanism of the inversion of the brain organoid structure, we implemented computational fluid dynamic (CFD) analysis both for orbital mixing and vertical mixing. Using CFD analysis, flow velocity, shear stress, strain rate, vorticity, turbulent energy, and energy dissipation were calculated in orbital mixing and three conditions of vertical mixing (45, 60, and 75 mm/s) (Fig. 3a). The turbulent energy presented a higher magnitude in vertical mixing than in orbital mixing, suggesting that the difference in turbulent energy may contribute to and be a positive factor for the formation of inverted brain organoids. We also calculated the velocity of each organoid using a solid–liquid mixed-phase flow analysis. The dispersion state in the orbital shaker is not equal, with movement of spheres along the container wall and toward the center (Fig. 3b, upper panel, Supplementary Video 1). Meanwhile, the organoids in vertical mixing are evenly dispersed throughout the entire culture tank (Fig. 3b, lower panel, Supplementary Video 2). Such flow characteristics and dispersion action might have a positive effect on the inverted brain structure by vertical mixing. Moreover, the discrete phase model (DPM) showed the difference in velocity of the organoids itself and the drag force applied to organoids between orbital mixing and vertical mixing. Analysis of three-dimensional flow velocity showed that organoids in orbital mixing rarely moved in the Z direction, while organoids in vertical mixing showed movement in the Z direction as well as X and Y directions (Fig. 3c). These results suggest that the formation of the inverted brain organoids obtained in this study might be due to the dispersive and leveling action of the organoids produced by vertical mixing and the low resistance that is applied to the movement of

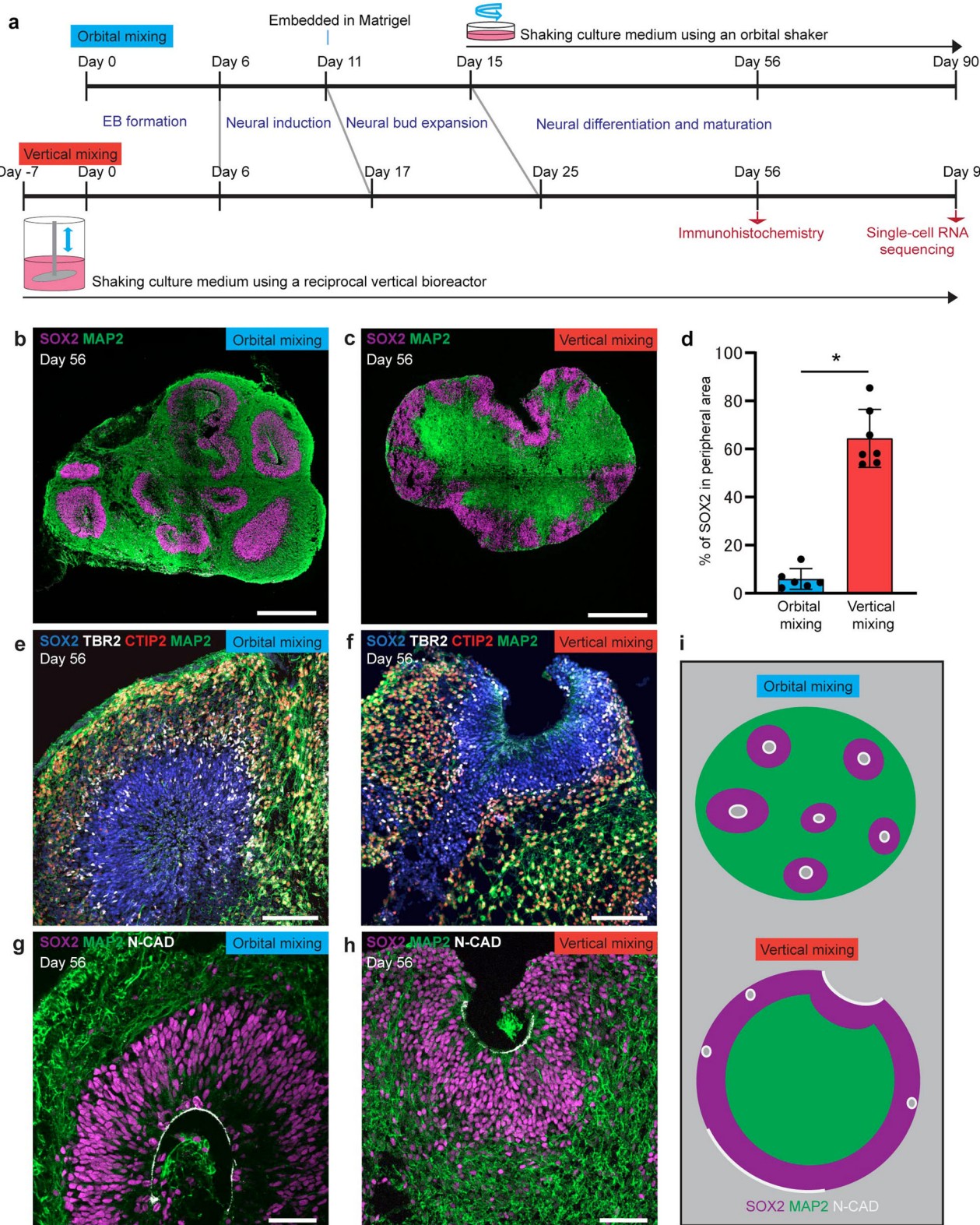

the organoids themselves. Besides, the turbulent energy of vertical mixing is higher than that of orbital mixing, suggesting the necessity of appropriate stimulation that the cells receive from the fluid in the induction of differentiation.

**Direction of primary in inverted brain organoids**. To reveal the link between fluid dynamics and cellular mechanosensing in vertical and orbital mixing, we performed immunohistochemical analysis of the primary cilium, the special sensory organelle in eukaryote cells, which receives signals from the environment and transduces them to the cell[19,20] and plays an important role in control proliferation, migration and neural patterning[19,20]. The primary cilium marker ARL13B, a small G protein localized in the cilia membrane, in SOX2-positive neural progenitors was investigated. Neuroepithelial cells lining the ventricle of

**Fig. 1 Brain organoid generated by vertical mixing showed inverted structure in comparison with brain organoid generated by orbital mixing.**
**a** Schematic diagram of conditions used to induce brain organoid by orbital mixing (upper schema) and by vertical mixing (lower schema).
**b**, **c** Immunostaining for neural progenitor (SOX2, magenta) and neuron (MAP2, green) in brain organoid generated by orbital mixing (**b**) or vertical mixing (**c**) on Day 56. **d** Quantification of SOX2-positive area in the peripheral region of brain organoid on Day 56. The peripheral region was defined as 100 μm inside from the edge of brain organoid. Brain organoid from vertical mixing showed higher percentage of SOX2-positive area in peripheral region in comparison with brain organoid from orbital mixing. Data represent mean ± SD ($n = 6$ for brain organoids by orbital mixing, $n = 7$ for brain organoids by vertical mixing). Difference between the two conditions was analyzed by Student's two-tailed *t*-test (*$p < 0.0001$). **e**, **f** Immunostaining for markers of neural progenitors (SOX2, blue), intermediate neural progenitors (TBR2, gray), and cortical neurons (CTIP2, red; MAP2, green) in brain organoids by orbital mixing (**e**) or vertical mixing (**f**) on Day 56. **g**, **h** Immunostaining for markers of ventricular neuroepithelial cells (N-CADHERIN: N-CAD, gray), neural progenitors (SOX2, magenta), and cortical neurons (MAP2, green) in brain organoids by orbital mixing (**g**) or vertical mixing (**h**) on Day 56. Note the apical side of organoids generated by orbital mixing is located inside organoids, while that of organoids generated by vertical mixing is located at the surface of organoids. **i** Schematic diagram shows the difference of brain organoids by orbital mixing and by vertical mixing. Scale bars = 500 μm (**b**, **c**), 100 μm (**e**, **f**), 50 μm (**g**, **h**).

orbital-mixing organoids also possess a primary cilium to the lumen of neural tube-like structures (Fig. 4a, Supplementary Fig. 3a, b). On the other hand, these cells in vertical-mixing organoids possess primary cilia toward the outer space of organoids (Fig. 4b, Supplementary Fig. 3c, d). We analyzed the direction of the primary cilium, calculated as the cilia angle between the cilia body and migration direction of neural progenitors, detected by co-staining with pericentrin, a protein localized at the centrosome (Fig. 4c, Supplementary Fig. 3e, f). While the primary cilia in neural progenitors of orbital-mixing organoids showed mainly two directions, those of vertical-mixing organoids showed random distribution in various directions (Fig. 4d). These data suggested that fluid dynamics in the bioreactors have a great impact on the inside-out or outside-in structure of iPSC-derived brain organoids via the direction of primary cilia in neural progenitor cells.

**Gene expression analysis of inverted brain organoids by single-cell RNA sequencing.** To analyze the cell types in inverted brain organoids, we performed single-cell RNA sequencing (scRNA-seq) of three organoids generated by orbital mixing and three organoids generated by vertical mixing on Day 90 of culture. Two thousand cells from each organoid were targeted, and in total 12,000 cells were analyzed. At first, we aligned and co-clustered the cells from organoids in both orbital and vertical mixing by Uniform Manifold Approximation and Projection (UMAP) algorithm and compared the expression of cells from each organoid (Fig. 5a). UMAP is the conventional dimensionality reduction of the data matrix of gene expressions in each cell. In the above process, cell clusters were defined using K-means clustering on principal component analysis (PCA) space, and the number of clusters was decided using the elbow method. Organoids in both orbital and vertical mixing enriched neuron marker-positive cells, and detailed distributions in UMAP were presented in Fig. 5b. Inverted brain organoids also presented markers for deeper layers of cortical neurons, such as *BCL11B* (also known as *CTIP2*), *TBR1*, and *SOX5*, and for upper-layer cortical neurons, such as *CUX2* and *SATB2* (Fig. 5b). Also, the expression of GABAergic marker genes (*DLX1*, *DLX5*, *GAD2*, and *NKX2.1*) was found in inverted brain organoids (Fig. 5b, Supplementary Fig. 4a), suggesting the induction of a number of GABAergic neurons via sonic hedgehog signaling[21,22]. Moreover, a heat map showed the differences in gene expression representing the typical brain region (Fig. 6a). These data suggest that brain organoids by orbital mixing have dorsal forebrain identity, while brain organoids by vertical mixing have ventral forebrain identity. We focused on the cilia-related signaling pathway in the population of SOX2-positive cells, based on our finding of the alteration of the cilia directions shown in Fig. 4. We analyzed scRNA-seq data of SOX2-positive cells in detail (Fig. 6b, Supplementary Tables 1, 2). The expression of

NKX2.1 was increased in SOX2-positive cells in vertical mixing organoids, which indicates the activation of sonic hedgehog signaling[21,22]. The cilia-related signal transduction is regulated by several factors associated with the dynamics of sonic hedgehog signaling[23]. SOX2-positive cells presented the alteration of gene expression to transduce cilia-related signaling such as GLI3, BOC, CDON, or GAS1[23,24] (Fig. 6b, Supplementary Tables 1, 2). Furthermore, the expressions of several genes related to cilia including CCDC88A[25] and DCX[26] were also altered (Fig. 6b, Supplementary Table 1, 2). These findings indicated that cilia-related signaling was affected by fluid dynamics in SOX2-positive cells. The function of primary cilia in controlling neuroepithelium polarity was reported in a previous study[27]. Therefore, we consider that primary cilia-related signaling is relevant to SOX2-positive cells, and that vertical mixing alters cilia-related signaling, leading to the structural changes of organoids in vertical mixing.

**Enrichment of GABAergic neurons in inverted brain organoids.** Immunostaining presented ventral neural progenitors on Day 56 and GABAergic neurons on Day 90 in both orbital mixing and vertical mixing, but with a predominant number in vertical mixing (Fig. 7a, b), consistent with the gene expression analysis (Figs. 5b, 6a, b, Supplementary Fig. 4a). Furthermore, we evaluated the generation of excitatory neurons and GABAergic neurons along the time axis in orbital mixing and vertical mixing, and found that vertical mixing might facilitate the promotion of GABAergic neuronal differentiation (Supplementary Fig. 5). These data suggested that the inverted brain organoid is a characteristic brain organoid that harbors a unique structure and special cell composition.

**Disease modeling by inverted brain organoids.** Several items of evidence have emerged to support the notion that alteration of GABAergic circuits contributes to Alzheimer's disease (AD) pathogenesis by disrupting the overall network function[28]. In order to utilize the inverted brain organoid for disease analysis, we generated inverted brain organoids from iPSCs derived from a healthy control subject and a familial AD patient carrying the deletion of E693 in APP protein, APP E693Δ[29]. After 2 months of culture in vertical mixing, brain organoids showed the inverted structure with the expression of hippocampal marker PROX1, consistent with a single-cell RNA-seq (Fig. 8, Supplementary Fig. 4b). Aβ oligomer accumulation was observed in MAP2-positive neurons of AD brain organoids (Fig. 8), suggesting that the inverted brain organoids could be a disease model in vitro.

**Discussion**
Brain organoids are a self-organization of three-dimensional aggregates resembling brain structures generated from human

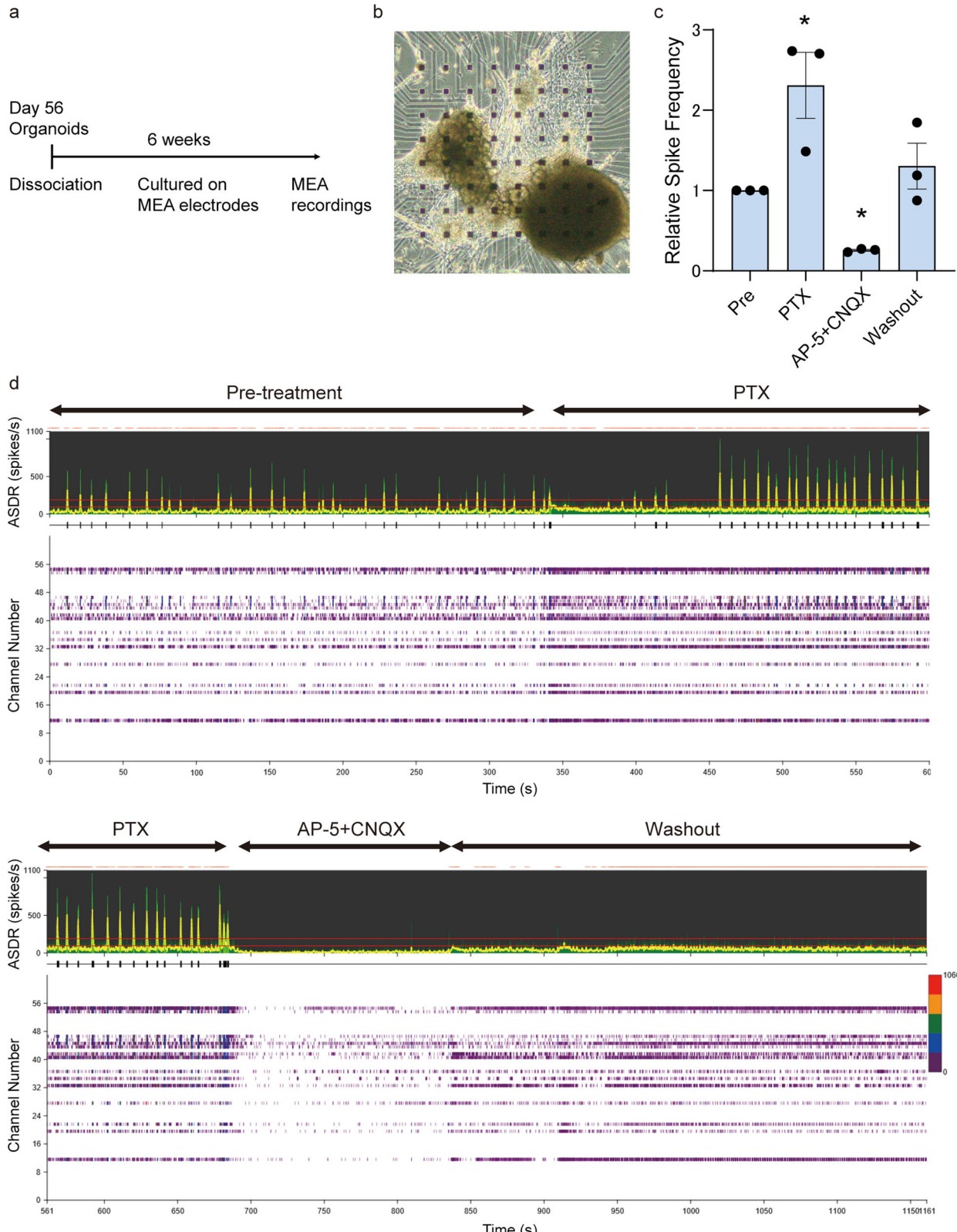

**Fig. 2 Inverted organoids exhibited functional properties. a** Timeline of the experiments. **b** A representative phase-contrast image of clumped organoids on MEA electrodes. **c** Quantification of spike frequency. Treatment with 100 μM PTX increased the spike frequency, and the addition of 50 μM AP-5 and 50 μM CNQX decreased it. ANOVA, $p < 0.05$, *post hoc $p < 0.05$, $n = 3$ organoids cultured on independent dishes. Pre: pre-treatment. **d** ASDR plots and individual raster plots for all 64 electrodes.

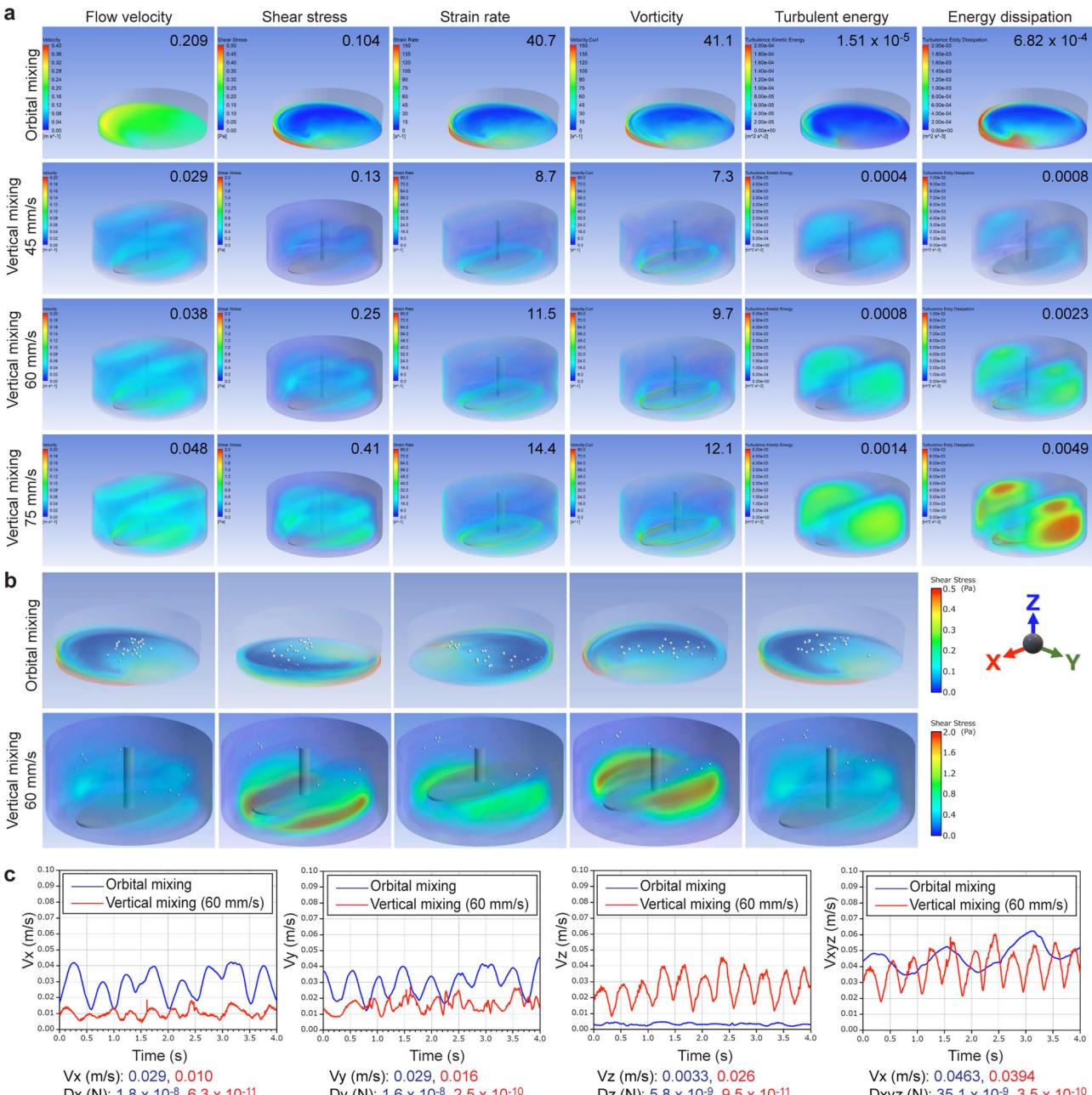

**Fig. 3 Vertical mixing has different physical characteristics from orbital mixing. a** Computational simulation of simple fluid dynamics to compare the rheological parameters during stirring by orbital mixing, or vertical mixing at a mixing speed of 45, 60, or 75 mm/s. The average value of each parameter is shown in the thumbnail of each panel. Flow velocity and shear stress of orbital mixing were higher than that of vertical mixing. Turbulent energy and energy dissipation, which are related to the status of the stirring force, of vertical mixing were higher than those of orbital mixing. **b** Computational simulation of solid–liquid transient analysis by using a discrete phase model to compare the movement of spheres (white circles) under orbital mixing or vertical mixing. Spheres in orbital mixing were unevenly distributed along with the wall of the culture dish, gradually translocated toward the center, and gathered with each other, while spheres in vertical mixing were equally dispersed throughout the culture bottle. **c** Comparison of three-dimensional flow velocity between orbital mixing (blue line) and vertical mixing (red line). The movement of spheres in orbital mixing showed higher velocity in X- and Y-direction than that in vertical mixing. In contrast, the movement of spheres only in vertical mixing showed rhythmic Z-directional velocity. The drag force applied to the spheres in orbital mixing is also higher than that in vertical mixing.

iPSCs. Although brain organoids recapitulate many key features of human brain development, the mechanism that controls the formation of brain organoids is still not fully understood. In this study, we produced inverted brain organoids that have a contradictory inside-out pattern of neural progenitors and neurons by the use of vertical mixing bioreactors. The fluid dynamics of the organoid and direction of a cellular mechanosensor, primary

cilia, in the outside layer of neural progenitor cells were different between vertical and orbital mixing. scRNA-seq analysis revealed that the inverted brain organoids contained a neuronal area of GABAergic neurons, and it was applicable to the analysis of neurological diseases.

In order to determine the mechanism involved in the generation of inverted brain organoids, we analyzed the physical factors

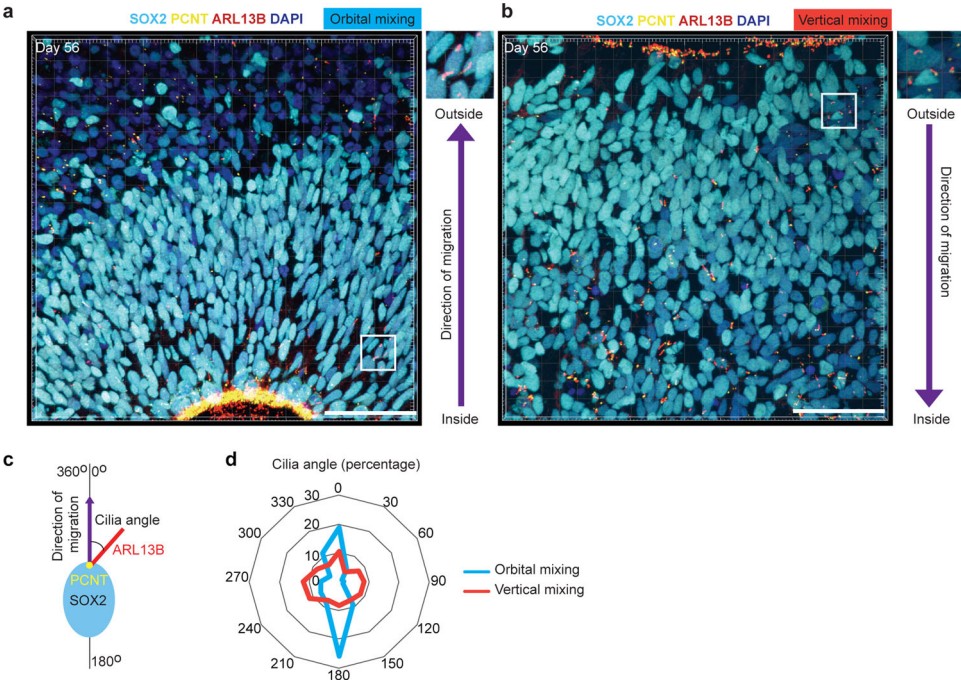

**Fig. 4 Primary cilia in neural progenitor cells were pointing in direction of migration in inverted brain organoid. a, b** Immunostaining for primary cilia (ARL13B, red) and cilia basal body (Pericentrin: PCNT, yellow) in ciliated SOX2-positive cells of brain organoid generated by orbital mixing or vertical mixing, respectively. The direction from the yellow dot (PCNT) to the red line (ARL13B) is an indicator of the direction of cilia in neural progenitor (SOX2, turquoise) (inset with higher magnification). Scale bars = 50 μm. **c** The method of measuring cilia angle. **d** Quantification of cilia angle based on the migration direction of ciliated SOX2-positive cells. Primary cilia of brain organoids generated by orbital organoid showed a bidirectional manner, whereas primary cilia of organoids generated by vertical organoid showed a multidirectional manner. Data were obtained from three different organoids with two or three different regions.

of fluid flow caused by vertical mixing. CFD analysis suggested that the turbulent energy of fluid flow might have a positive effect on the formation of inverted brain organoids. Moreover, the organoids in vertical mixing are equally dispersed throughout the culture tank due to the low drag force applied against them, suggesting that brain organoids in vertical mixing were floating freely in the low-stress flow of culture medium. Furthermore, the early cues may be imparted into iPSCs in the vertical mixing culture system that are retained long-term during their differentiation into brain organoids. On the other hand, the lack of extracellular matrix also might partially contribute to the inverted structure, although it was shown, in our study, that vertical mixing contributed to the generation of inverted brain structure. Previous studies used Matrigel, a source of many components of the extracellular matrix, as a scaffold for brain organoid growth and maturation[30]. Nevertheless, the random distribution of the primary cilia direction and the alteration of cilia-related signaling detected by scRNA-seq in neural progenitors by vertical mixing suggest that fluid dynamics control the signaling pathway that regulates neural patterning. It is of interest that primary cilia activity is essential for the establishment of apical-basal polarity of the radial glial scaffold and that early neuroepithelial deletion of ciliary ARL13B led to the reversal of the apical-basal polarity of radial progenitors[31].

Our scRNA-seq data showed that inverted brain organoids have alterations in gene expression. Suspension cultures of iPSCs in the bioreactor have been shown to regulate not only cell aggregation, but also gene expression[32]. This change might occur at the iPSCs stage and continue during the development of inverted brain organoids. We analyzed the characteristics of SOX2-positive cells from orbital mixing and vertical mixing by scRNA-seq and found that the gene expression patterns between

them were different. SOX2-positive cells from vertical mixing exhibited the increased expressions of GABAergic progenitor markers DLX2 and NKX2.1, as well as the GABAergic neuron marker GAD2. These findings suggested that SOX2-positive cells from vertical mixing harbored different characteristics from those of orbital mixing. Although SOX2 is a marker for neural progenitors, its expression has also been described in differentiated neurons in various regions of the nervous system[33], and SOX2 expression in GABAergic neurons in mice is also reported[34]. We consider that these references support our findings regarding the characteristics of SOX2-positive cells in organoids generated by vertical mixing. Furthermore, in the organoids generated by vertical mixing, the detection of GABAergic neurons by scRNA-seq with immunostaining is substantially increased in organoids generated by vertical mixing compared to that by orbital mixing. In the scRNA-seq data, we found altered gene expression that regulates cilia-related signaling in association with sonic hedgehog signaling in SOX2-positive cells. Previous studies have shown that abnormalities in cilia-related signaling can lead to changes in the migration of GABAergic neurons[23,35,36], suggesting an influence of cilia-related signaling in the development of GABAergic neurons. Therefore, we consider that alterations in cilia-related signaling with sonic hedgehog signaling activation may have contributed to the generation of GABAergic neurons. It is unclear whether the changes in cilia signaling-related genes are the cause or the result of the altered differentiation fate, but a previous report regarding a similar polarity of organoids shown in a midbrain organoid protocol with sonic hedgehog supported our findings[37].

In addition to the increase of GABAergic neurons, the results of single-cell analysis provided us with some insight regarding the fact that vertical mixing altered the composition of the cell

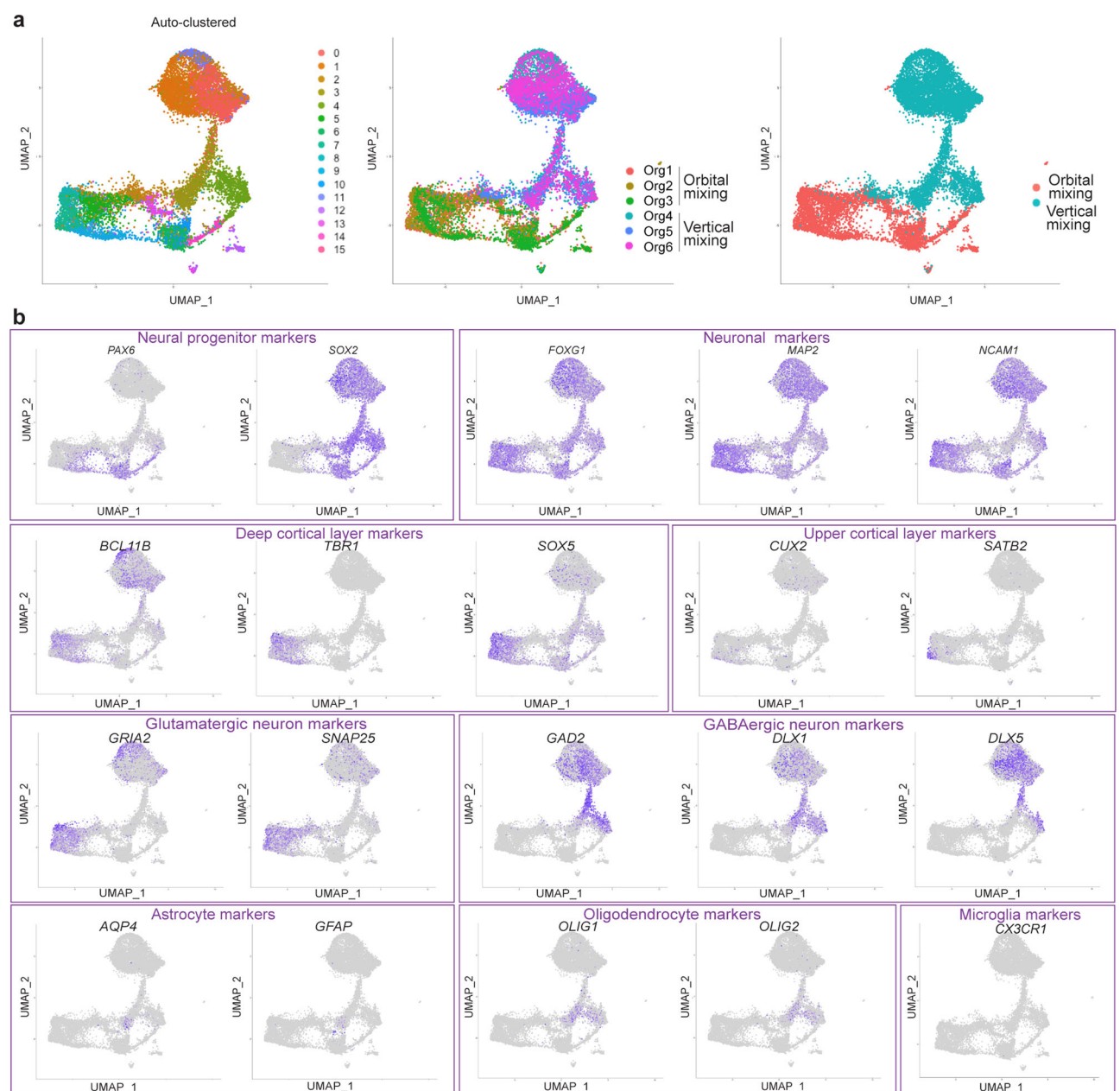

**Fig. 5 Single-cell RNA sequencing data presented differential gene expression between organoids generated by orbital mixing and vertical mixing.** **a** Dimension reduction and clustering (PCA-UMAP) in all cells of brain organoids generated by orbital mixing and vertical mixing. Three organoids were generated by orbital mixing: Org1, Org2, and Org3; three organoids were generated by vertical mixing: Org4, Org5, and Org6. **b** Major markers for neurons and glial cells were displayed in each panel. Neural progenitor markers: *PAX6* and *SOX2*; neuronal markers: *FOXG1*, *MAP2*, and *NCAM1*; deep cortical layer markers: *BCL11B* (or *CTIP2*), *TBR1*, and *SOX5*; upper cortical layer markers: *CUX2* and *SATB2*; glutamatergic neuron markers: *GRIA2* and *SNAP25*; GABAergic neuron markers: *GAD2*, *DLX1,* and *DLX5*; astrocyte markers: *AQP4* and *GFAP*; oligodendrocyte markers: *OLIG1* and *OLIG2*; microglia marker: *CX3CR1*.

population with cortical markers. There was a decrease in the number of deep cortical layer neurons in vertically mixed organoids as shown by the loss of TBR1 and SOX5. The relative increase in the number of GABAergic neurons resulted in a relative decrease in TBR1-positive cells, which are highly expressed in excitatory neurons[38], and SOX5-positive cells, which are specifically expressed in corticofugal neurons[39]. On the other hand, the number of BCL11B-positive cells was retained, as they represent certain subtypes of inhibitory neurons[40]. These alterations have adopted a ventral identity by vertical mixing. Taken together, the combination of physical factors might change the gene expression through the response of primary cilia, resulting in the changing neural patterning and differentiation fate in inverted brain organoids.

We analyzed a prominent feature of AD using inverted brain organoids. We are hopeful that these inverted brain organoids designed by controlling fluid dynamics will be a new model for studying human brain development and disorders.

## Methods

**Ethics statement**. Generation and use of human iPSC were approved by the Ethics Committee of each institute. All methods were performed in accordance with the approved guidelines. Formal informed consent was obtained from a subject.

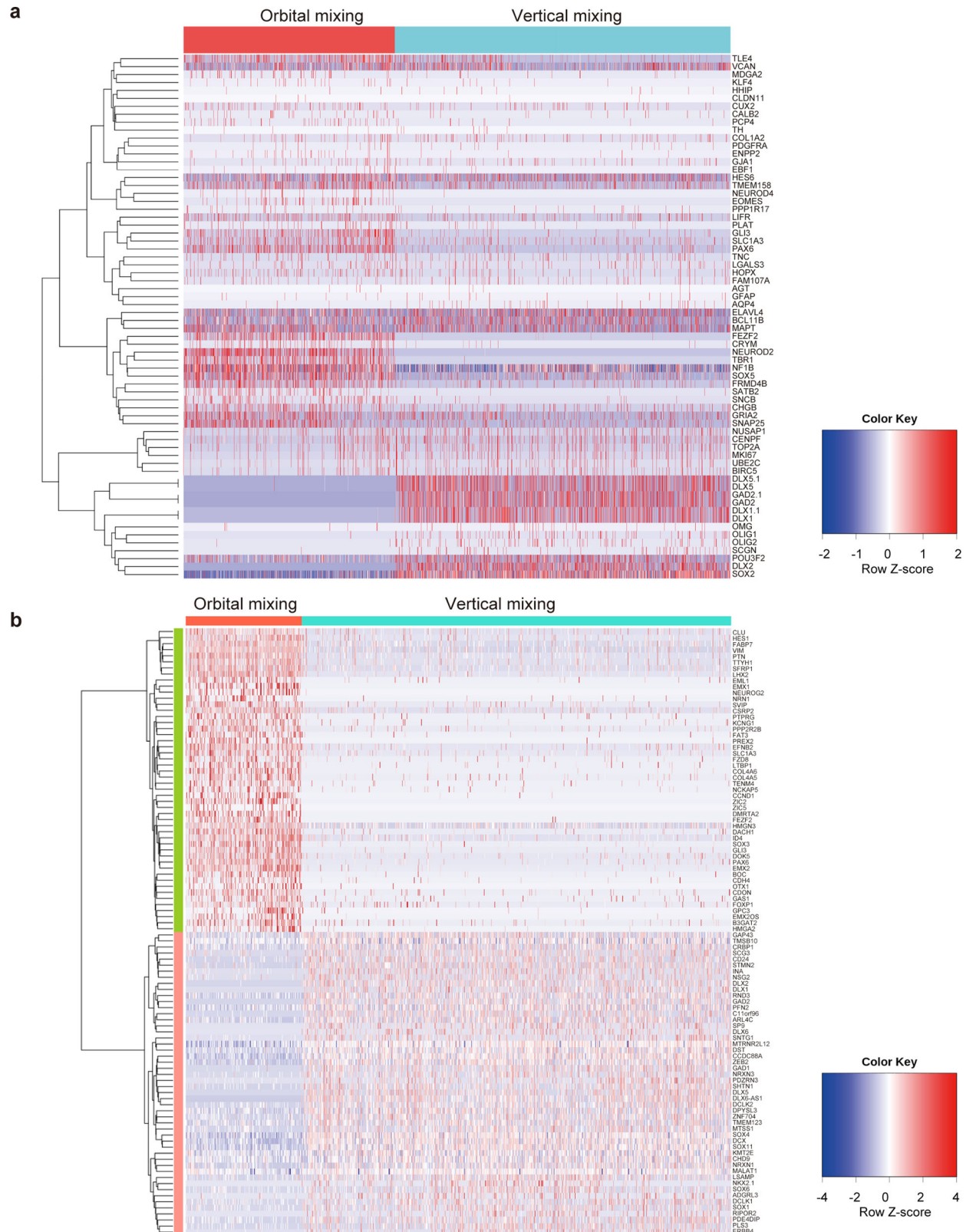

**Fig. 6 Gene expressions of organoids analyzed by scRNA-seq were altered between orbital mixing and vertical mixing. a** Heat map for brain lesion-related genes in total cells. **b** Heat map for genes altered in SOX2-positive cells.

**Maintenance of human iPSCs.** Human iPSCs were maintained on a recombinant fragment of laminin-511 (iMatrix-511™, Nippi, Tokyo, Japan) by using StemFit AK02N medium (Ajinomoto, Tokyo, Japan). We used 201B7 iPSC line derived from fibroblasts of a healthy subject[41] and APP1E111 iPSC line derived from fibroblasts of a patient with familial Alzheimer's disease[29].

**Generation of brain organoids by orbital mixing.** iPSCs were cultured up to 80% confluency (typically 10 days after passage). iPSC colonies were dissociated into single cells after 4 min of incubation in 0.5 x Tryple Select/0.25 mM EDTA (Thermo Fisher Scientific, Waltham, MA) at 37 °C and suspended in StemFit

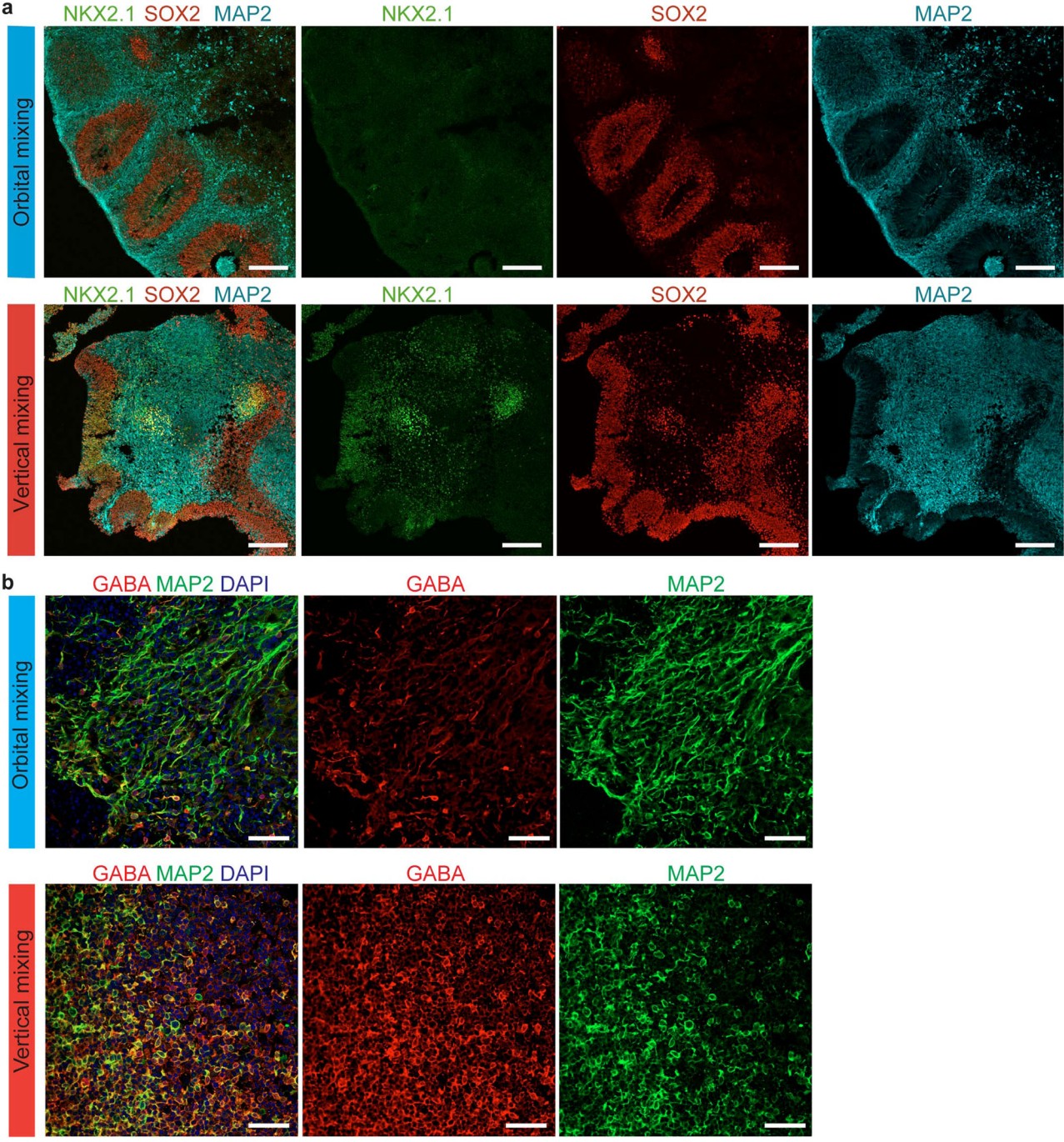

**Fig. 7 Inverted brain organoid presenting GABAergic neurons. a** Immunostaining with specific marker for ventral neural progenitor cells (NKX2.1, green) on Day 56. Organoids generated by vertical mixing exhibited NKX2.1-positive cells. Scale bars = 200 μm. **b** Immunostaining with specific markers for GABAergic neurons (GABA, red) and pan-neuron markers (MAP2, green) in brain organoid generated by orbital mixing and by vertical mixing on Day 90. Vertical mixing enriched GABA-positive cells. Scale bars = 50 μm.

AK02N with 10 μM of Y-27632 (Nacalai Tesque, Inc., Kyoto, Japan) to count the number and viability.

iPSCs were transferred into an embryoid body (EB) formation medium (EB medium) consisting of DMEM/F12 (Thermo Fisher Scientific) with 20% Knockout Serum replacement (Thermo Fisher Scientific), 3% Fetal bovine serum (FBS) (Thermo Fisher Scientific), 1% Glutamax (Thermo Fisher Scientific), 1% non-essential amino acids mix (NEAA, Thermo Fisher Scientific), 1% penicillin/streptomycin (Thermo Fisher Scientific), 4 ng/ml basic FGF (Wako Chemicals, Osaka, Japan), and 50 μM Y-27632. Dissociated 9000 alive iPSCs in EB medium were disseminated into single wells of U-bottom 96-well plates (ultra-low attachment type, NunclonTM SpheraTM microplates, 96U-Well Plate (174729), Thermo Fisher Scientific). The U-bottom 96-well plates were centrifuged at 200 × g for 3 min to make iPSCs aggregate quickly at the bottom of the well, and were kept in the incubator under a condition of 5% $CO_2$ at 37 °C. We defined the day of dissemination as Day 0. On Day 4, we replaced the medium with EB formation medium without basic FGF or Y-27632. On Day 6, we replaced the medium with neural induction medium consisting of DMEM/F12 with 1% N2 supplement (Thermo Fisher Scientific), 1% GlutaMAX, 1% NEAA, 1 μg/ml heparin (Nacalai Tesque, Inc.), and 1% penicillin/streptomycin. On Day 11, the outside of EBs became brighter and showed radial organization. EBs were transferred into cold droplets of MatrigelTM (Corning, Corning, NY) on a sheet of Parafilm (Parafilm® M PM996, Bermis, WI) with small 3-mm dimples in a 10-cm petri dish and were incubated for 20 min in an incubator at 37 °C to allow Matrigel polymerization. After the polymerization step, the EB-Matrigel droplets were removed from the Parafilm sheet and transferred into Neuroepithelial expansion medium, which consisted of a

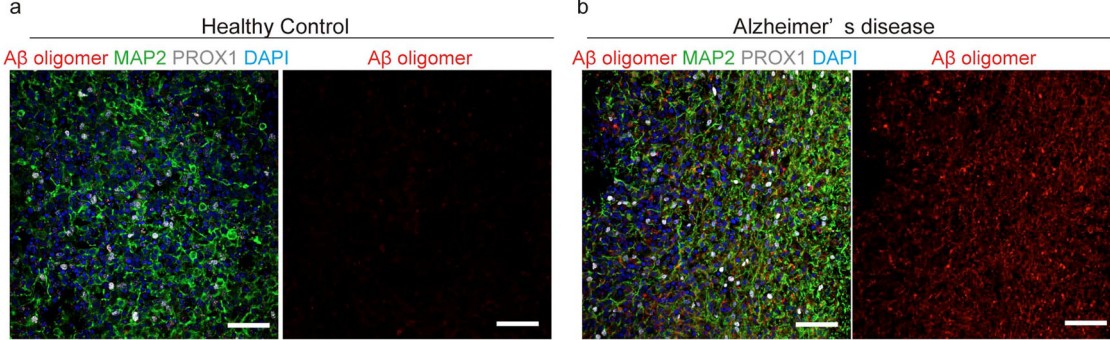

**Fig. 8 Analysis of Alzheimer's disease by using inverted brain organoids.** Images of Aβ oligomers in brain organoid of healthy control (**a**) and Alzheimer's disease with APP E693Δ mutation (**b**). Accumulation of Aβ oligomers was seen in Alzheimer's disease organoids. Scale bars = 50 μm.

1:1 mixture of DMEM/F12 and Neurobasal medium (Thermo Fisher Scientific) with 0.5% N2 supplement, 1% B-27 supplement without vitamin A Thermo Fisher Scientific), 1% GlutaMAX, 0.5% NEAA, insulin 2.5 µg/ml (I9278, Sigma), 1% penicillin/streptomycin, and 0.1% Amphotericin B (Thermo Fisher Scientific). On Day 15, 16 EB-Matrigel droplets were transferred into a 6-cm dish with 6 ml of Differentiation medium which consisting of 1:1 mixture of DMEM/F12 and Neurobasal medium with 0.5% N2 supplement, 1% B-27 supplement without AO (Thermo Fisher Scientific), 1% GlutaMAX, 0.5% NEAA, insulin 2.5 µg/ml, 1% penicillin/streptomycin, and 0.1% Amphotericin B. All 6-cm dishes were horizontally rotated on an orbital shaker (Cell Shaker, CS-LR 0081704-000, Taitec, Saitama, Japan), equipped inside the incubator, at a rotating speed of 60 rpm[42]. The culture medium was refreshed every 3 to 4 days.

From Day 40, EB-Matrigel droplets were cultured in differentiation medium with additional 1% Matrigel (growth factor reduced type, 354230, Corning). From Day 70, EB-Matrigel droplets were cultured in differentiation medium with additional 2% Matrigel (growth factor reduced type) and 2% B27 supplement without AO.

**Culture brain organoids by vertical mixing.** To establish vertical mixing under tight regulation of a stable temperature, pH, and dissolved $O_2$ concentration, we utilized the vertically mixing bioreactor system HiD 4×4 (Satake Co. Ltd, Tokyo, Japan), with the controlling system of cultivation condition, S-BOX×02 (Satake) (Supplementary Fig. 1).

iPSCs were cultured up to 80% confluence (typically 10 days after passage). iPSC colonies were dissociated into single cells after 4 min of incubation in 0.5 x Tryple Select/0.25 mM EDTA (Thermo Fisher Scientific) in a 37 °C incubator, and suspended in StemFit AK02N with 50 µM of Y-27632 (Nacalai Tesque, Inc.) for number counting and viability determination.

Dissociated $2.5 \times 10^6$ cells in 250 ml of StemFit AK02N plus 50 µM of Y-27632 were disseminated into a single-use bottle (Supplementary Fig. 1) specialized for suspension culture under continuous vertical mixing of the HiD 4×4 system (Satake) at a setting of 60 mm/s speed and 15-mm stroke with continuous air flow of 30 ml/s. After 7 days of cultivation in StemFit AK02N medium, disseminated iPSCs formed spheres of homogeneous size (~100–150-µm diameter), and the medium was refreshed with EB formation medium as described above. We defined the day of medium refreshment as Day 0. On Day 6, we replaced the medium with Neural induction medium, as described above. On Day 17, we replaced the medium with Neuroepithelial expansion medium, as described above. On Day 25, we replaced the medium with Differentiation medium, also as described above. Sphere cultivation in the vertical mixing system was continued up to Day 90, with weekly medium change under tight regulation of pH 6.5–7.5, $O_2$ concentration 20%, and temperature 32–34 °C.

**Histological and immunohistochemical analysis.** Tissues were fixed in 4% paraformaldehyde for 15 min (1-mm organoids) or 30 min (3-mm organoids) at room temperature, followed by three times washing in PBS for 10 min. Tissues were allowed to sink in 30% sucrose overnight and were then embedded in O.C.T. compound (Sakura Finetek, Tokyo, Japan) and quickly frozen in liquid $N_2$. Frozen tissues were cut into 12-µm slices by cryostat (CM1850, Leica Biosystems, Wetzlar, Germany) at −18 to −20 °C. For immunohistochemistry, sections were permeabilized in 0.5% Triton-X100/PBS (0.5% PBST) for 30 min at room temperature, and were then blocked in blocking solution consisting of 0.1% PBST with 10% normal donkey or goat serum for 2 h at room temperature. Sections were then incubated with primary antibodies in blocking solution at 4 °C overnight. These antibodies were used: anti-SOX2 (MAB2018, R&D System, 1:1000), anti-β3-Tubulin (D71G9, Cell Signaling Technology, 1:500), anti-MAP2 (ab5392, Abcam, 1:3000), anti-TBR2 (ab23345, Abcam, 1:1000), anti-CTIP2 (ab18645, Abcam, 1:1000), anti-N-cadherin (C3865, Sigma, 1:1000), anti-ARL13B (17711-1-AP, Proteintech, 1:400), anti-Pericentrin (ab28144, Abcam, 1:200), anti-SOX2 (14-9811-82, eBioscience, 1:400), anti-NKX2.1(MAB5460, Merk, 1:500), anti-GABA (A2052, Sigma, 1:300), anti-VGLUT1(135303, Synaptic Systems, 1:150), anti-

PROX1(ab199359, Abcam, 1:500), anti-Aβ oligomer specific antibody NU1[29,43] (1:500). After washing with 0.1% PBST four times, 15 min/time, samples were incubated with secondary antibodies of Alexa Flour 405/488/546/594 or 647 conjugates (Invitrogen, 1:1000) for 2 h at room temperature while protected from light. Then, samples were washed with 0.1% PBST four times, 15 min/time, and mounted with Prolong Gold mounting (ProLong™ Gold Antifade Mountant, Thermo Fisher Scientific) and kept stable in the dark at room temperature for about 20 h. Data were observed under fluorescent microscopy (confocal FV1000, Olympus or IN Cell analyzer 6000, GE Healthcare).

ImageJ software (NIH, USA) was used to measure the percentage of SOX2-positive cells in the peripheral area of brain organoids and the direction of primary cilia.

**Computational fluid dynamic analysis.** Physical flow simulations in a cell culture dish/bottle were performed using CFD software, and ANSYS Fluent 2019 R3 (ANSYS, Inc.) was used for calculation. For verification of the turbulence model, we performed the Realizable κ-ε model. For gas–liquid boundary tracking method with the cell culture dish, we used the VOF model. For liquid boundary tracking method with a bottle, we used the Slip wall boundary condition.

For calculating the drag force, we used the following equation:

$$D = \frac{1}{2} \rho_f V^2 S C_D$$

$$C_D = \begin{cases} \frac{24}{Re} & (Re < 2) \\ \frac{10}{\sqrt{Re}} & (2 < Re < 500) \\ 0.44 & (500 < Re < 10^5) \end{cases}$$

$$Re = \frac{\rho_f u d}{\mu_f}$$

D: drag (N); $\rho_f$: fluid density (kg/m³); $\mu_f$: fluid viscosity (Pa-s); V: Relative velocity of fluid and particle (m/s); S: Cross-section of particle (m²); $C_D$: Drag coefficient (–); Re: Reynolds number (–); u: particle movement speed (m/s); d: particle size (m).

**MEA recordings.** The organoids were each incubated with accumax for 15 min at 37 °C and dissociated with slow pipetting to 2–3 clumps, and cell mixtures from independent individual organoids including clumps and dissociated cells were cultured on 64-channel MEA chips (MED-R515A; Alpha Med Scientific) coated with Polyethyleneimine (Sigma) and Laminin-511 (Nippi) at 37 °C in 5% $CO_2$. For culture on MEAs, cell mixture from one organoid was plated on electrodes of a dish as a droplet with 100 µl of medium for 2 h, and then 1 ml of medium was added. Half of the medium was changed every 2 days. After 6 weeks, spontaneous extracellular field potentials were acquired using a 64-channel MEA system (MED64-Basic; Alpha Med Scientific).

**Single-cell RNA sequencing.** Dissociated cells were resuspended in PBS containing 1% BSA, immediately followed by a library preparation targeting single cells using the Chromium Single Cell 3′ Reagent Kit v3 (10×Genomics) and Chromium Controller (10×Genomics) according to the manufacturer's instructions. Three organoids generated by orbital mixing and three organoids generated by vertical mixing were analyzed. Two thousand cells from each organoid were targeted, and a total of 12,000 cells were analyzed. The library was sequenced on a HiSeq2500 TruSeq SBS v3 reagent. Cell-specific FASTQ files were generated by deconvolution of UMIs and cell barcodes using bcl2fastq 2.20.0.422 (Illumina). Alignment to the human reference genome GRCh38 and UMI counting were conducted by Cell Ranger v3.1.0 pipeline (10×Genomics). Uniform manifold approximation and projection (UMAP) implemented in the Seurat package v3.2.3 was conducted using

1st to 10th principal components after filtering out the cells with a high number of detected genes (≥9000) and RNA molecules (≥60,000), and with a high percentage of mitochondrial genes (>60%). The differentially expressed genes were identified using the Wilcoxon rank-sum test and heat map was plotted using the pheatmap R package version 1.0.12.

**Statistics and reproducibility**. Results were analyzed using student's two-tailed *t*-test or one-way ANOVA followed by Dunnett's post hoc test to determine statistical significances of the data. Differences were considered significant at $p < 0.05$. Analyses were performed using GraphPad Prism software version 8.0 (GraphPad Software, San Diego, CA).

**Reporting summary**. Further information on research design is available in the Nature Research Reporting Summary linked to this article.

## Data availability

The single-cell RNA-seq data has been deposited at NCBI/BioProject under accession number GSE184409. Source data underlying all figures have been provided as Supplementary Data 1. All other data that support the findings of this study are available from the corresponding author upon reasonable request.

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

## Acknowledgements

We would like to express our sincere gratitude to all of our co-workers and collaborators: Kayoko Tsukita, Takako Enami, and Ayako Nagahashi for their technical support, Mikie Iijima, Nozomi Kawabata, Tomomi Urai, and Miho Nagata for their valuable administrative support, and Minako Tateno for critical reading of the manuscript. This research was funded in part by a grant for Core Center for iPS Cell Research of the Research Center Network for Realization of Regenerative Medicine from AMED (H.I.) and the Uehara Memorial Foundation (H.I.) and partially supported by the Center of Innovation Program from MEXT and JST (H.I.).

## Author contributions

H.I. conceived the project. DN.AS., K.I., I.I., R.K., S.S., T.O., Y.K., T.K., Y.Y. and A.W. performed the experiments. W.L.K. provided antibodies. DN.AS., K.I., T.K. and H.I. wrote the manuscript.

## Competing interests

The authors declare no competing interests.
