## [Peer Review File · Communications Biology]

Reviewers' comments:

Reviewer #2 (Remarks to the Author):

It has been well-known that mechanical forces can affect stem cell differentiation, but it is the "how" that has always been elusive. In this manuscript, Suong et al generated brain organoids from human iPSCs cultured in a vertical mixing system, and characterized the organoids to compare with organoids generated from conventional orbital shaking cultures. They reported that unlike orbital organoids forming self-contained neural tube-like structures inside of the organoids, the vertical organoids form neuroepithelium layers at the periphery of the organoid, establishing an inverted outside-in structure. These inverted organoids have their outer surface as the apical surface, while neurons are born and migrate into the interior of the organoids. Using computational fluid dynamics, the authors calculated the differences of mechanical forces exerted on the organoids in the two culture systems. Single-cell RNA sequencing showed the two culture systems generated organoids with different cell type composition and changes in expression of gene pathways, including changes in genes related to primary cilium signaling, which is complemented by changes in primary cilium orientation analyzed by immunostaining. In addition, inverted organoids also contain more GABAergic neuron. Finally, the author gave a brief proof of principle to show that the inverted organoid derived from patients can be used as a disease model. The writing of the manuscript is clear and easy to follow. The data presented are of good quality and figures are well-organized. The use of computational fluid dynamics to quantitatively analyze the differences of the culture systems brings interesting insight that is often neglected by stem cell biologists like myself. However, a striking problem in the organoid protocols used by the authors fundamentally confounds the interpretation of the results presented in the study, particularly regarding to whether the inverted culture is responsible for the formation of the inverted structures. As the authors themselves briefly acknowledged in the Discussion, the inverted organoid protocol skipped the Matrigel embedding step employed in the orbital organoid protocol, and the lack of extracellular matrix in the inverted organoids may be the determining factor for the resulting structure. I find it shocking that the authors have omitted to control for this key experimental variable in their organoid culture, and continued to compare the resulting organoids. The extracellular matrix protein contained in Matrigel mimics the basement membrane of in vivo neurodevelopment. From various published organoid methods to our lab's in-house experience of developing several brain region-specific organoid protocols, the use of Matrigel to embed or coat the organoid surface allows the neuroepithelium cells to recognize the outer surface as the basal surface, thus forming the inside-out neural tube-like structures with apical surface closing into a lumen inside. In the case of this study, I do believe that the lack of Matrigel is responsible for the inverted structure formation than the vertical culture system. The authors should control variables and embed the EBs in the vertical culture into Matrigel the same way as for the orbital organoids, or alternatively, remove the Matrigel step for the orbital organoids to draw comparison. This fundamental flaw in the experimental design must be addressed before this manuscript is ready for publication.

Below are a few additional comments and suggestions that the authors should consider to address:

- 1) The MEA experiments in Fig.2 is only performed for the inverted organoids. I'm curious how they would compare to orbital organoids, and if inverted organoids offer any advantages.
- 2) The immunostaining images in Fig. 4a,b will be more clear if images for the individual channels is shown separately.
- 3) In the scRNAseq results in Fig.5, SOX2, a definitive neural progenitor cell marker is extensively overlapping and co-expressing with neuronal markers such as MAP2, as well as GABAergic neuron marker GAD2. This is very surprising. Are these cells neurons or neural progenitors? Or if the inverted organoids contain a new cell type that exhibit signatures of both cell types?
- 4) The authors should characterize the presence of ventral neural progenitor cell marker NKX2.1 via both immunostaining and scRNAseq in the inverted organoids.
- 5) The argument that inverted organoids have ventral forebrain identity (line 150) is contradictory with the wide-spread presence of CTIP2+ cortical neurons found in the dorsal forebrain. From the data presented, the inverted organoids seem to contain a heterogeneous mixture of regions representing dorsal forebrain and regions representing ventral. Thus, the statement in line 176, "the inverted brain organoids inside a presented homogeneous neuronal area were enriched with GABAergic neurons" is incorrect.
- 6) The authors provided no evidence suggesting a link between primary cilia signaling to

neuroepithelium polarity before rashly jumping into the conclusion that “fluid dynamics in the bioreactors have a great impact on the inside-out or outside-in structure of iPSC derived brain organoids via the direction of primary cilium in neural progenitor cells.” (Line 134).

Reviewer #3 (Remarks to the Author):

The research study by Suong et al. describes a novel methodology to induce inverted morphological structures in brain organoids via a vertical-mixing technology. Conventional approaches in generating brain organoids have largely been dependent on orbital shaking methods and it is currently unknown how mechanical forces affect brain organoid development. This study is the first to demonstrate that mechanical forces applied through vertical mixing can have potent effects on the development of brain organoids. Interestingly, brain organoids generated by this method produce an inverted developmental program in organoids as neural progenitors are observed lining the outer perimeter and neurons are observed within the center. The authors provide evidence that the primary cilium in neural progenitor cells may play a role for this inverted organoid morphology as they were shown to have a random distribution of cilia angles compared to bi-directional cilia angles in organoids generated via orbital shaking. In addition, a marked increased level of GABAergic neurons are produced in vertically-mixed brain organoids compared to organoids which were generated via orbital shaking. While my enthusiasm for the findings of this study remain high, I do have a significant number of concerns that if addressed would provide for an excellent contribution to Communications Biology.

Major Concerns:

- 1.) A fundamental mechanistic understanding into how brain organoids self-organize into three-dimensional structures with morphological features similar to the human brain is important in developing improved organoid protocols. In addition, uncovering principles that control human brain development may be important in understanding neurological disease mechanisms. The authors provide data supporting that vertical mixing of organoid cultures result in an inverted arrangement of neural progenitors on the periphery and neurons in the center of organoids. However, it is unclear whether inverted organoids generated by this approach was solely due to vertical mixing or due to earlier steps within their protocol that combined vertical mixing in conjunction with earlier steps in their protocol as shown in Figure 1a (i.e. iPSC culture, EB formation, neural bud expansion). In conventional protocols, orbital shaking initiates after these earlier steps and it would be more clear to interpret the authors findings if they initiated vertical mixing after these earlier steps as well. For example, if vertical mixing were initiated at D15 (similar to orbital shaking), then would inverted organoids form? To more directly determine the effects of vertical mixing on organoid development, it is important that the authors initiate the orbital shaking and vertical mixing at similar time points in organoid development.
- 2.) The current study is lacking critical details regarding the number of technical and biological replicates used for various experiments. For example, how many independent experiments were performed? How many organoids were analyzed for IHC, single cell RNA sequencing, cilia angle measurements, and MEA recordings? In addition, please provide statistical analyses for the single cell RNA sequencing experiments. 2,000 cells were analyzed for sc-RNA sequencing, which appears low for these types of analyses.
- 3.) The authors provide evidence that the percentage of GABAergic neurons is substantially higher in vertically-mixed organoids compared to organoids generated via orbital shaking. Could the authors comment how mechanical forces could potentially alter neurodevelopmental trajectories in organoids. Is there literature to support this association?

Minor Concerns:

- 1.) I would suggest rephrasing the title of the study as it does not clearly convey the advancement to the field. I would suggest the following or something similar this is more concise: “ Induction of inverted morphology in brain organoids by vertical-mixing bioreactors.”

- 2.) There are several dozen instances of spelling/grammatical and word choice errors throughout the manuscript. For example, line 28 reads "In the organoids by vertical-mixing, developing neurons migrated from the outer..." This sentence should be rephrased as the following: "Organoids generated by vertical-mixing showed neurons that migrated from the outer..."
- a. Line 52: "various portions of brain and discrete cell types....". This sentence should be rephrased as the following: "various portions of the brain and discrete cell types....".
 - b. Line 130: "between cilia body and migration direction..." should read "between the cilia body and migration direction..."
 - c. Line 169: "three-dimensional aggregates generated from..." should read "three-dimensional aggregates resembling brain structures generated from..."
 - d. Line 201 "using inverted brain organoid" should read "inverted brain organoids."
- 3.) In the Introduction, please include a few sentences that compare and contrast vertical-mixing with orbital shaking methodologies. Please also reference other recent advances in the field analyzing the effects of mechanical forces on organoid development:
- e. Montes-Olivas et al: Mathematical Models of Organoid Cultures.
 - f. Gota-Silva et al: Computational fluid dynamic analysis of physical forces playing a role in brain organoid cultures in two different multiplex platforms.
 - g. Dahl-Jensen et al: The physics of organoids: a biophysical approach to understanding organogenesis.
- 4.) What effect does vertical mixing have on the growth trajectories of organoids? Are organoids generated via vertical mixing the same size as organoids generated by orbital shaking? Could the authors provide data for these observations.
- 5.) Based on the data provided by the computational fluid dynamic analysis, vertically mixed organoids show greater level of turbulent energy compared to orbital shaking. Using conventional orbital shaking methods, cells within the organoid core undergo cell death as organoids become larger. It would bring increased value to this study if the authors provided data showing the levels of cell death using cleaved caspase-3 staining as a marker in vertically-mixed organoids compared to organoids generated by orbital shaking.
- 6.) In Figure 2, could the authors provide more details in the "Materials and Methods" how these experiments were performed. In line 89, the authors dissociate brain organoids into clumps and then were cultured on MEA chips. Please define clumps and specify procedures how they were prepared. Did the authors also evaluate electrophysiology from organoids generated by orbital shaking? How do they compare? In addition, could the authors please specify how many organoids were tested and how many MEA chips (technical replicates) utilized for these experiments.
- 7.) In Figure 4, could the authors please provide larger pictures of the higher power inset within panels A and B. Please also overlay schematic C within these higher power insets to more clearly show how ciliary angles are measured.
- 8.) In line 201, the authors state they modeled some aspects of AD using inverted brain organoids, however they only assessed A-beta oligomers. Please modify the text accordingly. For example, line 201 could read similar to the following: "We modeled a prominent feature of AD using inverted brain organoids."

Response to reviewers' comments:

Reviewers' comments:

Reviewer #2 (Remarks to the Author):

It has been well-known that mechanical forces can affect stem cell differentiation, but it is the “how” that has always been elusive. In this manuscript, Suong et al generated brain organoids from human iPSCs cultured in a vertical mixing system, and characterized the organoids to compare with organoids generated from conventional orbital shaking cultures. They reported that unlike orbital organoids forming self-contained neural tube-like structures inside of the organoids, the vertical organoids form neuroepithelium layers at the periphery of the organoid, establishing an inverted outside-in structure. These inverted organoids have their outer surface as the apical surface, while neurons are born and migrate into the interior of the organoids. Using computational fluid dynamics, the authors calculated the differences of mechanical forces exerted on the organoids in the two culture systems. Single-cell RNA sequencing showed the two culture systems generated organoids with different cell type composition and changes in expression of gene pathways, including changes in genes related to primary cilium signaling, which is complemented by changes in primary cilium orientation analyzed by immunostaining. In addition, inverted organoids also contain more GABAergic neuron. Finally, the author gave a brief proof of principle to show that the inverted organoid derived from patients can be used as a disease model.

The writing of the manuscript is clear and easy to follow. The data presented are of good quality and figures are well-organized. The use of computational fluid dynamics to quantitatively analyze the differences of the culture systems brings interesting insight that is often neglected by stem cell biologists like myself.

However, a striking problem in the organoid protocols used by the authors fundamentally confounds the interpretation of the results presented in the study, particularly regarding to whether the inverted culture is responsible for the formation of the inverted structures. As the authors themselves briefly acknowledged in the Discussion, the inverted organoid protocol skipped the Matrigel embedding step employed in the orbital organoid protocol, and the lack of extracellular matrix in the inverted organoids may be the determining factor for the resulting structure. I find it shocking that the authors have omitted to control for this key experimental variable in their organoid culture, and continued to compare the resulting organoids. The extracellular matrix protein contained in Matrigel mimics the basement membrane of in vivo neurodevelopment. From various published organoid methods to our lab's in-house experience of developing several brain region-specific organoid protocols, the use of Matrigel to embed or coat the organoid surface allows the neuroepithelium cells to recognize the outer surface as the basal surface, thus forming the inside-out neural tube-like structures with apical surface closing into a lumen inside. In the case of this study, I do believe that the lack of Matrigel is responsible for the inverted structure formation than the vertical culture system. The authors should control variables and embed the EBs in the vertical culture into Matrigel the same way as for the orbital organoids, or alternatively, remove the Matrigel step for the orbital organoids to draw comparison. This fundamental flaw in the experimental design must be addressed before this manuscript is ready for publication.

>We greatly appreciate Reviewer #2's constructive comments. We conducted additional experiments to investigate the effect of Matrigel by removing the embed step of Matrigel for the orbital organoids, and found that the removal of Matrigel did not promote the generation

of inverted brain. The % of SOX2-positive area in the peripheral region, which is defined as being within 100 μm from the edge of brain organoids, was $8.4 \pm 6.7\%$ ($n = 4$ organoids, mean \pm SD). On the other hand, those of orbital mixing organoids and vertical mixing organoids were $5.9 \pm 4.3\%$ ($n = 6$) and $64.4 \pm 12.1\%$ ($n = 7$), respectively, as presented in Figure 1d. Thus, we consider that vertical mixing mainly contributes to the production of inverted brain, although it can not be completely ruled out that removal of Matrigel makes a contribution to the inversion. We added the data in Supplementary Figure 2 and the description in Results, as below.

Supplementary Figure 2

Figure legends

Organoids were generated by orbital mixing without Matrigel. Generated organoids were analyzed by immunostaining with SOX2 and MAP2, and did not present the inverted pattern. The % of SOX2-positive area in the peripheral region of brain organoids was $8.4 \pm 6.7\%$ ($n = 4$ organoids, mean \pm SD), not showing the increase of inverted ratio. Scale bars =500 μm .

Results:

Page 4, Line 90 in highlighted version of the manuscript,

We investigated the effect of Matrigel for organoid structure by removing the Matrigel embed step for the orbital organoids, and found that the removal of Matrigel did not promote the generation of inverted structure (Fig.S2). Therefore, we consider that vertical mixing mainly contributes to the production of inverted brain structure, although removal of Matrigel may contribute to the inversion.

(Reviewer #2 Comment 1)

1) The MEA experiments in Fig.2 is only performed for the inverted organoids. I'm curious how they would compare to orbital organoids, and if inverted organoids offer any advantages.

>We are grateful for Reviewer #2's comment. We conducted MEA experiments using organoids generated by orbital mixing and vertical mixing. We investigated whether vertically mixing organoids exhibited functional properties to clarify the validity of organoids generated by current methods. We confirmed that vertical organoids exhibited the functionality with compound responsibility.

Regarding the advantage of inverted organoids, thanks to the reviewer's comment, we could find the generation of GABAergic neurons or progenitor cells expressing specific markers including NKX2.1, DLX2, and GAD2, as below. However, it was difficult to compare and describe the advantage by MEA data in vitro due to the technical limitations.

(Reviewer #2 Comment 2)

2) The immunostaining images in Fig. 4a, b will be more clear if images for the individual channels is shown separately.

>We thank Reviewer #2 for this comment. We added data on the individual channels to Supplementary Figure 3 according to the suggestion.

Figure legend
The immunostaining and analysis of cilium

(a,c) SOX2 staining, (b,d) PTCN and ARL13B staining, (e,f) measurement of cilia directions. Scale bar = 20 μ m.

(Reviewer #2 Comment 3)

3) In the scRNAseq results in Fig.5, SOX2, a definitive neural progenitor cell marker is extensively overlapping and co-expressing with neuronal markers such as MAP2, as well as GABAergic neuron marker GAD2. This is very surprising. Are these cells neurons or neural progenitors? Or if the inverted organoids contain a new cell type that exhibit signatures of both cell types?

>We greatly appreciate Reviewer #2's comments.

We reanalyzed the characteristics of SOX2-positive cells from orbital mixing and vertical mixing by single-cell RNAseq, and we found that the gene expression patterns between them were different. SOX2-positive cells from vertical mixing exhibited the increased expression of GABAergic progenitor markers NKX2.1 and DLX2, and GABAergic neuron marker GAD2. These findings suggested that SOX2-positive cells from vertical mixing harbored different characteristics from those of orbital mixing. Although SOX2 is a marker for neural progenitors, it has been reported that its expression has also been described in differentiated neurons in various regions of the nervous system (Int. J. Mol. Sci. 2019, 20, 4540), and SOX2 expression in GABAergic neurons in mice had been reported (Development. 2008, 135:541-557). We consider that these references may support our findings regarding the characteristics of SOX2-positive cells in organoids generated by vertical mixing showing the inclusion of GABAergic progenitors and GABAergic neurons. We added the single-cell RNAseq data in Figure 6b and Supplementary Table 1,2, as below.

Figure legend

(b) Heat map for genes altered in SOX2-positive cells.

Discussion:

Page 9, Line 219, in highlighted version of the manuscript,

We analyzed the characteristics of SOX2-positive cells from orbital mixing and vertical mixing by scRNA-seq and found that the gene expression patterns between them were different. SOX2-positive cells from vertical mixing exhibited the increased expressions of GABAergic progenitor markers NKX2.1 and DLX2, as well as the GABAergic neuron marker GAD2. These findings suggested that SOX2-positive cells from vertical mixing harbored different characteristics from those of orbital mixing. Although SOX2 is a marker for neural progenitors, its expression has also been described in differentiated neurons in various regions of the nervous system³³, and SOX2 expression in GABAergic neurons in mice is also reported³⁴. We consider that these references support our findings regarding the characteristics of SOX2-positive cells in organoids generated by vertical mixing.

(Reviewer #2 Comment 4)

4) The authors should characterize the presence of ventral neural progenitor cell marker NKX2.1 via both immunostaining and scRNA seq in the inverted organoids.

>We greatly appreciate Reviewer #2's comments. We conducted additional analysis of NKX2.1 expression by single-cell RNAseq and immunostaining, and characterized the presence of ventral neural progenitor cell marker NKX2.1 via both immunostaining and single-cell RNAseq in the inverted organoids. We added the data in Figure 7a and Supplementary Figure 3, as below.

Figure 7a legend

(a) Immunostaining with specific marker for ventral neural progenitor cell (NKX2.1, cyan) on Day 56. Organoids generated by vertical mixing exhibited NKX2.1-positive cells. Scale bars = 200 μ m.

Figure legend of Supplementary Figure 4.

Single-cell RNAseq analysis demonstrated that NKX2.1-positive cells were detected predominantly in organoids generated by vertical mixing.

(Reviewer #2 Comment 5)

5) The argument that inverted organoids have ventral forebrain identity (line 150) is contradictory with the wide-spread presence of CTIP2+ cortical neurons found in the dorsal forebrain. From the data presented, the inverted organoids seem to contain a heterogeneous mixture of regions representing dorsal forebrain and regions representing ventral. Thus, the statement in line 176, “the inverted brain organoids inside a presented homogeneous neuronal area were enriched with GABAergic neurons” is incorrect.

>We greatly appreciate Reviewer #2’s comment. We revised our description as below.

Page 8, Line 194, in highlighted version of the manuscript,

scRNA-seq analysis revealed that the inverted brain organoids contained ~~homogeneous~~ a neuronal area ~~enriched~~ of GABAergic neurons, and it was applicable to the analysis of neurological diseases modeling. ~~Single-cell RNA sequencing analysis revealed that the inverted brain organoids inside a presented homogeneous neuronal area were enriched with GABAergic neurons and were applicable to neurological disease modeling.~~

(Reviewer #2 Comment 6)

6) The authors provided no evidence suggesting a link between primary cilia signaling to neuroepithelium polarity before rashly jumping into the conclusion that “fluid dynamics in the bioreactors have a great impact on the inside-out or outside-in structure of iPSC derived brain organoids via the direction of primary cilium in neural progenitor cells.” (Line 134).

>We are grateful for Reviewer #2’s comment. We analyzed single-cell RNAseq data of SOX2-positive cells in detail. The expression of NKX.2.1 was increased in SOX2-positive cells in vertical mixing organoids, which indicates the activation of sonic hedgehog signaling (Development 2002, 129, 4963-74, Cereb Cox 2006, 16,89-95). The cilia signal transduction is regulated by several factors associated with the dynamics of sonic hedgehog signaling (Front Cell Dev Biol 9, 630161). SOX2-positive cells presented the alteration of gene expression to transduce cilia-related signaling such as GLI3, BOC, CDON, or GAS1. Furthermore, expression of several genes related to cilia including CCDC88A, DCX, EMX1, or EMX2 were also altered. These findings indicated that cilia-related signaling was affected by fluid dynamics in SOX2-positive cells.

The function of primary cilia in controlling neuroepithelium polarity was reported in a previous study (Development 2012, 139:95-105). Therefore, we consider that primary cilia signaling is relevant to neuroepithelium polarity.

We re-analysed and revised Figure 6b as below, and added the description in the Discussion.

Figure legend

(b) Heat map for genes showing differences in SOX2-positive cells.

Results:

Page 7, Line 161, in highlighted version of the manuscript,

We analyzed scRNA-seq data of SOX2-positive cells in detail (Fig. 6b, Supplementary Table 1, 2). The expression of NKX2.1 was increased in SOX2-positive cells in vertical mixing organoids, which indicates the activation of sonic hedgehog signaling^{21,22}. The cilia-related signal transduction is regulated by several factors associated with the dynamics of sonic hedgehog signaling²³. SOX2-positive cells presented the alteration of gene expression to transduce cilia-related signaling such as GLI3, BOC, CDON, or GAS1^{23,24} (Fig. 6b, Supplementary Tables 1, 2). Furthermore, the expressions of several genes related to cilia including CCDC88A²⁵ and DCX²⁶ were also altered (Fig. 6b, Supplementary table 1, 2). These findings indicated that cilia-related signaling was affected by fluid dynamics in SOX2-positive cells. The function of primary cilia in controlling neuroepithelium polarity was reported in a previous study²⁷. Therefore, we consider that primary cilia-related signaling is relevant to SOX2-positive cells, and that vertical mixing alters cilia-related signaling, leading to the structural changes of organoids in vertical mixing.

Reviewer #3 (Remarks to the Author):

The research study by Suong et al. describes a novel methodology to induce inverted morphological structures in brain organoids via a vertical-mixing technology. Conventional approaches in generating brain organoids have largely been dependent on orbital shaking methods and it is currently unknown how mechanical forces affect brain organoid development. This study is the first to demonstrate that mechanical forces applied through vertical mixing can have potent effects on the development of brain organoids. Interestingly, brain organoids generated by this method produce an inverted developmental program in organoids as neural progenitors are observed lining the outer perimeter and neurons are observed within the center. The authors provide evidence that the primary cilium in neural progenitor cells may play a role for this inverted organoid morphology as they were shown to have a random distribution of cilia angles compared to bi-directional cilia angles in organoids generated via orbital shaking. In addition, a marked increased level of GABAergic neurons are produced in vertically-mixed brain organoids compared to organoids which were generated via orbital shaking. While my enthusiasm for the findings of this study remain high, I do have a significant number of concerns that if addressed would provide for an excellent contribution to Communications Biology.

>We greatly appreciate reviewer #3's positive and constructive comments.

Major Concerns:

(Reviewer #3 Comment 1)

1) A fundamental mechanistic understanding into how brain organoids self-organize into three-dimensional structures with morphological features similar to the human brain is important in developing improved organoid protocols. In addition, uncovering principles that control human brain development may be important in understanding neurological disease mechanisms. The authors provide data supporting that vertical mixing of organoid cultures result in an inverted arrangement of neural progenitors on the periphery and neurons in the center of organoids. However, it is unclear whether inverted organoids generated by this approach was solely due to vertical mixing or due to earlier steps within their protocol that combined vertical mixing in conjunction with earlier steps in their protocol as shown in Figure 1a (i.e. iPSC culture, EB formation, neural bud expansion). In conventional protocols, orbital shaking initiates after these earlier steps and it would be more clear to interpret the authors findings if they initiated vertical mixing after these earlier steps as well. For example, if vertical mixing were initiated at D15 (similar to orbital shaking), then would inverted organoids form? To more directly determine the effects of vertical mixing on organoid development, it is important that the authors initiate the orbital shaking and vertical mixing at similar time points in organoid development.

>We are very grateful for Reviewer #3's comments. To directly determine the effects of vertical mixing on organoid development, we conducted additional experiments to transfer organoids to vertical mixing from orbital mixing or to orbital mixing from vertical mixing on Day 15. In the experiments of organoids being transferred to vertical mixing from orbital mixing on Day 15, all organoids were broken probably due to vulnerability against mechanical stress, and they could not be cultured over 40 days in the repeated experiments, as below.

Phase images on Day 40. Scale bar = 2 mm.

On the other hand, we found that inverted organoids were generated when orbital mixing was initiated on Day 15 with medium change for orbital mixing on Day 12 following vertical mixing, as below. The % of SOX2-positive area in the peripheral region, which is defined as being within 100 μm from the edge of brain organoids, was $46.8 \pm 25.4\%$ ($n = 3$ organoids, mean \pm SD). Therefore, we consider that the time point to initiate the vertical mixing would be crucial for the generation of inverted organoids, although we could not test many possibilities to address Reviewer #3's comment.

On Day 56. Scale bar = 1,000 μm .

Inverted organoids were generated when orbital mixing was initiated on Day 15 following vertical mixing. The % of SOX2-positive area in the peripheral region, which is defined as within 100 μm from the edge of brain organoids, was $46.8 \pm 25.4\%$ ($n = 3$ organoids, mean \pm SD).

(Reviewer #3 Comment 2)

2) The current study is lacking critical details regarding the number of technical and biological replicates used for various experiments. For example, how many independent experiments were performed? How many organoids were analyzed for IHC, single cell RNA sequencing, cilia angle measurements, and MEA recordings? In addition, please provide statistical analyses for the single cell RNA sequencing experiments. 2,000 cells were analyzed for sc-RNA sequencing, which appears low for these types of analyses.

>We are grateful for Reviewer #3's comments. We added the information of the experiments below. We described the number of organoids analyzed by immunostaining, single-cell RNA sequencing, and MEA recordings in each of the figure legends.

We analyzed a total of 12,000 cells by single-cell RNAseq. Three organoids generated by orbital mixing and three organoids generated by vertical mixing were analyzed. 2,000 cells from each organoid were targeted, and in total 12,000 cells were analyzed. We apologize for the confusion. We revised and added the description in Methods and Results, as below.

Methods

Page 15, Line 390, in highlighted version of the manuscript,

Dissociated cells were resuspended in PBS containing 1% BSA, immediately followed by a library preparation targeting **single2,000** cells using the Chromium Single Cell 3' Reagent Kit v3 (10×Genomics) and Chromium Controller (10×Genomics) according to the manufacturer's instructions. **Three organoids generated by orbital mixing and three organoids generated by vertical mixing were analyzed. 2,000 cells from each organoid were targeted, and in total 12,000 cells were analyzed.** The library was sequenced on a HiSeq2500 TruSeq SBS v3 reagent. Cell-specific FASTQ files were generated by deconvolution of UMIs and cell barcodes using bcl2fastq 2.20.0.422 (Illumina). Alignment to the human reference genome GRCh38 and UMI counting were conducted by Cell **Ranger v3.1.0 pipeline (10×Genomics)**. **Uniform manifold approximation and projection (UMAP) implemented in the Seurat package v3.2.3 was conducted using 1st to 10th principal components after filtering out the cells with a high number of detected genes ($\geq 9,000$) and RNA molecules ($\geq 60,000$), and with a high percentage of mitochondrial genes ($> 60\%$).** The differentially expressed genes were identified using the Wilcoxon rank-sum test and a heatmap was plotted using the pheatmap R package version 1.0.12.

Page 6, Line 144, in highlighted version of the manuscript,

Results

To analyze the cell types in inverted brain organoids, we performed single-cell RNA sequencing (scRNA-seq) of **three organoids generated by orbital mixing and three organoids generated by vertical mixing** on Day 90 of culture. **Two thousand cells from each organoid were targeted, and in total 12,000 cells were analyzed.**

(Reviewer #3 Comment 3)

3) The authors provide evidence that the percentage of GABAergic neurons is substantially higher in vertically-mixed organoids compared to organoids generated via orbital shaking. Could the authors comment how mechanical forces could potentially alter neurodevelopmental trajectories in organoids. Is there literature to support this association?

>We greatly appreciate Reviewer #3's comments. From the results of RNAseq analysis, we found altered gene expression that regulates cilia-related signaling in association with sonic hedgehog signaling in SOX2-positive cells. Previous studies have shown that abnormalities in cilia-related signaling can lead to changes in the migration of GABAergic neurons, suggesting the influence of cilia signaling in the development of GABAergic neurons.

Therefore, we consider that alterations in cilia signaling related to sonic hedgehog signaling activation may have contributed to the formation of GABAergic neurons in vertical mixing. We added the explanation in the Discussion, as below.

Discussion:

Page 9, line 228, in highlighted version of the manuscript,

Furthermore, in the organoids generated by vertical mixing, the detection of GABAergic neurons by scRNA-seq with immunostaining is substantially increased in organoids generated by vertical mixing compared to that by orbital mixing. In the scRNA-seq data, we found altered gene expression that regulates cilia-related signaling in association with sonic hedgehog signaling in SOX2-positive cells. Previous studies have shown that abnormalities in cilia-related signaling can lead to changes in the migration of GABAergic neurons^{23,35,36}, suggesting an influence of cilia-related signaling in the development of GABAergic neurons. Therefore, we consider that alterations in cilia-related signaling with sonic hedgehog signaling activation may have contributed to the formation of GABAergic neurons.

Minor Concerns:

(Reviewer #3 Comment 4)

1) I would suggest rephrasing the title of the study as it does not clearly convey the advancement to the field. I would suggest the following or something similar this is more concise: “Induction of inverted morphology in brain organoids by vertical-mixing bioreactors.”

>We greatly appreciate Reviewer #3’s comments. We revised the title to “Induction of inverted morphology in brain organoids by vertical-mixing bioreactors” as Reviewer #3 suggested.

(Reviewer #3 Comment 5)

2) There are several dozen instances of spelling/grammatical and word choice errors throughout the manuscript. For example, line 28 reads “In the organoids by vertical-mixing, developing neurons migrated from the outer...” This sentence should be rephrased as the following: “Organoids generated by vertical-mixing showed neurons that migrated from the outer...”

a. Line 52: “various portions of brain and discrete cell types....”. This sentence should be rephrased as the following: “various portions of the brain and discrete cell types....”.

b. Line 130: “between cilia body and migration direction...” should read “between the cilia body and migration direction...”

c. Line 169: “three-dimensional aggregates generated from...” should read “three-dimensional aggregates resembling brain structures generated from...”

d. Line 201” “using inverted brain organoid” should read “inverted brain organoids.”

>We are grateful for Reviewer #3’s comments. We revised the manuscript as below.

Page 2 Line 29, in the highlighted version of the manuscript,

The organoids **generated** by vertical-mixing **showed neurons that** migrated from the outer periphery to the inner core of organoids, in contrast to orbital mixing.

Page 3, Line 52, in highlighted version of the manuscript,

Brain organoids from human iPSCs retain these outstanding advantages in preparing various portions of **the** brain and discrete cell types with neuronal circuitry³⁻¹⁰, and a variety of brain organoid protocols have been developed.

Page 6, Line 137, in highlighted version of the manuscript,

We analyzed the direction of the primary cilium, calculated as the cilia angle between **the** cilia body and migration direction of neural progenitors, detected by co-staining with pericentrin, a protein localized at the centrosome.

Page 8, Line 188, in highlighted version of the manuscript,

Brain organoids are a self-organization of three-dimensional aggregates **resembling brain structures** generated from human iPSCs. Although brain organoids recapitulate many key features of human brain development, the mechanism that controls the formation of brain organoids is still not fully understood.

Page 10, Line 240, in highlighted version of the manuscript,

We analyzed a prominent feature of AD using inverted brain organoids. AD inverted brain organoids showed widespread accumulation of A β oligomers in neurons.

(Reviewer #3 Comment 6)

3) In the Introduction, please include a few sentences that compare and contrast vertical-mixing with orbital shaking methodologies. Please also reference other recent advances in the field analyzing the effects of mechanical forces on organoid development:

e. Montes-Olivas et al:

f. Gota-Silva et al: Computational fluid dynamic analysis of physical forces playing a role in brain organoid cultures in two different multiplex platforms.

g. Dahl-Jensen et al: The physics of organoids: a biophysical approach to understanding organogenesis.

>We are thankful for Reviewer #3's comments. We added the information in the Introduction, as below.

Introduction:

Page 3, Line 58, in highlighted version of the manuscript,

Organoids are cultured as three-dimensional cell aggregates in floating condition using shakers or bioreactors^{1,15}, and various mechanical stimuli including shear stress and turbulent energy, as well as energy dissipation affect cell differentiation and organoid formation. There have been some studies regarding the effects of mechanical forces on organoid development¹⁶⁻¹⁸.

References;

16 Dahl-Jensen, S. & Grapin-Botton, A. The physics of organoids: a biophysical approach to understanding organogenesis. *Development* **144**, 946-951, doi:10.1242/dev.143693 (2017).

17 Goto-Silva, L. *et al.* Computational fluid dynamic analysis of physical forces playing a role in brain organoid cultures in two different multiplex platforms. *BMC Dev Biol* **19**, 3, doi:10.1186/s12861-019-0183-y (2019).

18 Montes-Olivas, S., Marucci, L. & Homer, M. Mathematical Models of Organoid Cultures. *Front Genet* **10**, 873, doi:10.3389/fgene.2019.00873 (2019).

(Reviewer #3 Comment 7)

4) What effect does vertical mixing have on the growth trajectories of organoids? Are

organoids generated via vertical mixing the same size as organoids generated by orbital shaking? Could the authors provide data for these observations.

>We greatly appreciate Reviewer #3's comments. We conducted additional experiments and presented the data of growth trajectories of organoids as below. The organoids generated by vertical mixing presented smaller size compared to those by orbital mixing, but they showed similar growth trajectories to those of orbital organoids.

In each experiment, 30 organoids were evaluated. Data were presented as Mean \pm SD by experimental triplication.

(Reviewer #3 Comment 8)

5) Based on the data provided by the computational fluid dynamic analysis, vertically mixed organoids show greater level of turbulent energy compared to orbital shaking. Using conventional orbital shaking methods, cells within the organoid core undergo cell death as organoids become larger. It would bring increased value to this study if the authors provided data showing the levels of cell death using cleaved caspase-3 staining as a marker in vertically-mixed organoids compared to organoids generated by orbital shaking.

>We greatly appreciate this comment from Reviewer #3. We conducted additional experiments to evaluate cleaved caspase-3-positive cells, and we found that both organoids generated by orbital mixing and by vertical mixing contained cleaved caspase-3-positive cells, as presented below.

(Reviewer #3 Comment 9)

6) In Figure 2, could the authors provide more details in the “Materials and Methods” how these experiments were performed. In line 89, the authors dissociate brain organoids into clumps and then were cultured on MEA chips. Please define clumps and specify procedures how they were prepared. Did the authors also evaluate electrophysiology from organoids generated by orbital shaking? How do they compare? In addition, could the authors please specify how many organoids were tested and how many MEA chips (technical replicates) utilized for these experiments.

>We greatly appreciate Reviewer #3’s comments. We described the details in Methods as below. We conducted MEA experiments using organoids generated by orbital mixing and vertical mixing. We investigated whether vertical mixing organoids exhibited functional properties to clarify the validity of organoids generated by the current methods, and thus we did not compare the MEA data between organoids by orbital mixing and vertical mixing. We conducted MEA experiments of 3 organoids cultured on independent dishes. We also added the description in the legend of Figure 2.

Results:

Page 15, Line 390, in highlighted version of the manuscript,
MEA recordings

The organoids were each incubated with accumax for 15 minutes at 37°C and dissociated with slow pipetting to 2-3 clumps, and cell mixtures from independent individual organoids including clumps and dissociated cells were cultured on 64-channel MEA chips (MED-R515A; Alpha Med Scientific) coated with Polyethyleneimine (Sigma) and Laminin-511 (Nippi) at 37°C in 5% CO₂. For culture on MEAs, cell mixture clumps from one organoid was plated on electrodes of a dish as a droplet with 100 µl of medium for 2 hours, and then 1 ml of medium was added. Half of the medium was changed every two days. After 6 weeks, spontaneous extracellular field potentials were acquired using a 64-channel MEA system (MED64-Basic; Alpha Med Scientific).

Figure legend of Figure 2

(c) Quantification of spike frequency. Treatment with 100 µM PTX increased the spike frequency, and the addition of 50 µM AP-5 and 50 µM CNQX decreased it. ANOVA, $p < 0.05$, *post hoc $p < 0.05$, $n = 3$ organoids cultured on independent dishes.

(Reviewer #3 Comment 10)

7) In Figure 4, could the authors please provide larger pictures of the higher power inset within panels A and B. Please also overlay schematic C within these higher power insets to more clearly show how ciliary angles are measured.

>We greatly appreciate this comment by Reviewer #3. We added the data in the Supplementary Figure as below.

Figure legend

The immunostaining and analysis of cilium

(a,c) SOX2 staining, (b,d) PTCN and ARL13B staining, (e,f) measurement of cilia directions. Scale bar = 20 μ m.

(Reviewer #3 Comment 11)

8) In line 201, the authors state they modeled some aspects of AD using inverted brain organoids, however they only assessed A-beta oligomers. Please modify the text accordingly. For example, line 201 could read similar to the following: “We modeled a prominent feature of AD using inverted brain organoids.”

>We greatly appreciate Reviewer #3’s comments. We revised the sentence as below.

Page 8, Line 194, in highlighted version of the manuscript, scRNA-seq analysis revealed that the inverted brain organoids inside the presented ~~homogeneous~~ neuronal area were shown enriched with GABAergic neurons and were applicable for analyzing neurological diseases modeling.

Page 10, Line 240, in highlighted version of the manuscript,
We ~~analyzed modeled some aspects of~~ a prominent feature of AD using inverted brain organoids.

Reviewers' comments:

Reviewer #1 (Remarks to the Author):

In the revised manuscript, Inoue and colleagues have addressed majority of the concerns I raised in the previous review with additional experiment results. Most importantly, the outstanding concern that the inverted morphology was caused by the removal of Matrigel embedding step has been dismissed with direct comparison. Overall, the revised manuscript is convincing in showing how changes in mechanical signaling can significantly alter the differentiation fate of brain organoids, independent of the biochemical cues supplied in the culture media.

It is particularly interesting that the vertical mixing leads to alternation in differentiation fate that resembles a process of ventralization, as now shown by the dramatically elevated expression of NKX2.1+ ventral progenitor and GABAergic neurons. Similar inverted polarity of organoids was previously seen in a midbrain organoid protocol that supplemented sonic hedgehog(shh) in orbital culture(Qian et al., 2016). The link between primary cilia signaling to neuroepithelium polarity and differentiation fate remains weak, but I agree with the authors that it may be topic for a separate study. It is unclear whether the changes in cilia signaling-related genes is the cause or the result of the altered differentiation fate.

In Fig 5, at first, I found it surprising that there is a significant overlap between BCL11B (CTIP2) and GABAergic neuron markers, because CTIP2 is known as a classic marker for deep layer excitatory neurons in the cerebral cortex (dorsal forebrain). However, I did a quick search for BCL11B on the human adult MTG data from Allen Institute and it indeed shows BCL11B expression in some inhibitory neuron clusters (<https://celltypes.brain-map.org/rnaseq/human/mtg>).

Therefore, I think the statement in Line 155, "Inverted brain organoids also presented deeper layers of cortical neurons, specific for BCL11B+" is incorrect. The likely explanation is that the entire inverted organoid has adopted a ventral identity and the BCL11B+ cells represent certain subtype(s) of inhibitory neurons. It will be helpful for the authors to comment more on their identities with more a thorough literature search.

After addressing this one remaining issue, I think the revised manuscript will be in a very improved shape and I recommend its publication in Communications Biology.

Qian, X., Nguyen, H.N., Song, M.M., Hadiono, C., Ogden, S.C., Hammack, C., Yao, B., Hamersky, G.R., Jacob, F., Zhong, C., et al. (2016). Brain-Region-Specific Organoids Using Mini-bioreactors for Modeling ZIKV Exposure. *Cell* 165, 1238-1254.

Reviewer #2 (Remarks to the Author):

I have reviewed the revised manuscript by Suong et al and believe the authors have not adequately addressed the points raised by the first review, and therefore I cannot recommend this manuscript in its current form to be published in Communications Biology. Although substantial revisions and experimentation have been provided by the authors, the inclusion of new data in the revised manuscript brings forth new questions and reduces the overall impact of this manuscript.

For example, my first major concern (comment 1) relates to the conclusions drawn from the vertical mixing methodology. To better compare orbital shaking with vertical mixing methodologies, I suggested a more direct experimental plan that would initiate both types of mixing methodologies initiating at day 15. The new data provided by the authors suggests that the effects of vertical mixing imparts its main effects in inducing inverted morphology in organoids at some timepoint between Day -7 and Day 15. This interpretation is based on the author's data showing that shifting organoid culture from vertical mixing to orbital shaking on Day 15 is not statistically significant from continued vertical mixing based on the % of SOX2+ cells in the peripheral area of the organoid. This new data is important in understanding the impact of vertical mixing on the directed differentiation of iPSCs into brain organoids and should be included in the revised manuscript. The author's data suggests that early cues are imparted into iPSCs that are retained long term during their differentiation into brain organoids.

Additionally, I have concerns relevant to the sc-RNA seq data as presented in Figure 5. As shown

in panel A, there are 15 individual clusters, but it is unclear what cell types these clusters represent. It appears the author's data was analyzed with a high level of cellular resolution, which by itself is not necessarily incorrect, but it becomes more important to then specifically annotate each cell cluster. Based on panel B, the author's data suggests that there is a loss of deep cortical layer neurons in vertically mixed organoids as shown by loss of TBR1 and SOX5. Could the authors please comment on this result? Another interpretation of their sc-data is that organoids generated vertical mixing alters the timing of neurodevelopment. Since GABAergic neurons develop at a later time point than excitatory neurons, it is possible that vertical mixing accelerates the timing of neurodevelopment in organoids. Could the authors provide more immunohistochemical analyses of vertical mixed organoids at earlier time points?

Lastly, the authors suggest their novel methodology to generate organoids via vertical mixing could be used to model neurological disorders. Although I agree that the inverted morphological phenomenon induced in organoids is highly interesting and can be used to understand basic principles of human brain development, I am skeptical that this model could be applied to modeling neurological disorders since the inverted morphology phenotype is not physiological.

Response to Reviewers' comments

Reviewers' comments:

Reviewer #1

In the revised manuscript, Inoue and colleagues have addressed majority of the concerns I raised in the previous review with additional experiment results. Most importantly, the outstanding concern that the inverted morphology was caused by the removal of Matrigel embedding step has been dismissed with direct comparison. Overall, the revised manuscript is convincing in showing how changes in mechanical signaling can significantly alter the differentiation fate of brain organoids, independent of the biochemical cues supplied in the culture media.

It is particularly interesting that the vertical mixing leads to alternation in differentiation fate that resembles a process of ventralization, as now shown by the dramatically elevated expression of NKX2.1+ ventral progenitor and GABAergic neurons. Similar inverted polarity of organoids was previously seen in a midbrain organoid protocol that supplemented sonic hedgehog(shh) in orbital culture (Qian et al., 2016). The link between primary cilia signaling to neuroepithelium polarity and differentiation fate remains weak, but I agree with the authors that it may be topic for a separate study. It is unclear whether the changes in cilia signaling-related genes is the cause or the result of the altered differentiation fate.

In Fig 5, at first, I found it surprising that there is a significant overlap between BCL11B (CTIP2) and GABAergic neuron markers, because CTIP2 is known as a classic marker for deep layer excitatory neurons in the cerebral cortex (dorsal forebrain). However, I did a quick search for BCL11B on the human adult MTG data from Allen Institute and it indeed shows BCL11B expression in some inhibitory neuron clusters (<https://celltypes.brain-map.org/rnaseq/human/mtg>). Therefore, I think the statement in Line 155, "Inverted brain organoids also presented deeper layers of cortical neurons, specific for BCL11B+" is incorrect. The likely explanation is that the entire inverted organoid has adopted a ventral identity and the BCL11B+ cells represent certain subtype(s) of inhibitory neurons. It will be helpful for the authors to comment more on their identities with more a thorough literature search.

After addressing this one remaining issue, I think the revised manuscript will be in a very improved shape and I recommend its publication in Communications Biology.

Response to Reviewer #1's comments

>We greatly appreciate reviewer#1's constructive comments. According to Reviewer #1's

comments, the following three points have been addressed.

1. We added the description and the citation regarding the similar inverted polarity of organoids that was previously seen in a midbrain organoid protocol supplemented sonic hedgehog(shh) in orbital culture.

Discussion, Page 8, Line 235 in highlighted version of the manuscript;

It is unclear whether the changes in cilia signaling-related genes are the cause or the result of the altered differentiation fate, but a previous report regarding a similar inverted polarity of organoids shown in a midbrain organoid protocol with sonic hedgehog supported our findings³⁷. Ref 37. Qian, X. *et al.* Brain-Region-Specific Organoids Using Mini-bioreactors for Modeling ZIKV Exposure. *Cell* **165**, 1238-1254, doi:10.1016/j.cell.2016.04.032 (2016).

2. We revised the description “Inverted brain organoids also presented deeper layers of cortical neurons, specific for BCL11B+” as below.

Results, Page 6, line 151 in highlighted version of the manuscript;

Inverted brain organoids also presented markers for deeper layers of cortical neurons, such as *BCL11B* (also called *CTIP2*), *TBR1* and *SOX5*, and for upper layer cortical neurons, such as *CUX2* and *SATB2* (Fig. 5b).

3. We added the description of “the entire inverted organoid has adopted a ventral identity and the BCL11B-positive cells represent certain subtypes of inhibitory neurons” with citation of the reference as below.

Discussion Page 10, Line 245 in highlighted version of the manuscript;

On the other hand, the number of BCL11B-positive cells was retained, as they represent certain subtypes of inhibitory neurons⁴⁰. These alterations have adopted a ventral identity by vertical mixing.

Ref 40. Nikouei, K., Muñoz-Manchado, A. B. & Hjerling-Leffler, J. BCL11B/CTIP2 is highly expressed in GABAergic interneurons of the mouse somatosensory cortex. *J Chem Neuroanat* **71**, 1-5, doi:10.1016/j.jchemneu.2015.12.004 (2016).

Reviewer #2 comments:

1) I have reviewed the revised manuscript by Suong et al and believe the authors have not adequately addressed the points raised by the first review, and therefore I cannot recommend this manuscript in its current form to be published in Communications Biology. Although substantial revisions and experimentation have been provided by the authors, the inclusion of new data in the revised manuscript brings forth new questions and reduces the overall impact of this manuscript.

For example, my first major concern (comment 1) relates to the conclusions drawn from the vertical mixing methodology. To better compare orbital shaking with vertical mixing methodologies, I suggested a more direct experimental plan that would initiate both types of mixing methodologies initiating at day 15. The new data provided by the authors suggests that the effects of vertical mixing imparts its main effects in inducing inverted morphology in organoids at some timepoint between Day -7 and Day 15. This interpretation is based on the author's data showing that shifting organoid culture from vertical mixing to orbital shaking on Day 15 is not statistically significant from continued vertical mixing based on the % of SOX2+ cells in the peripheral area of the organoid. This new data is important in understanding the impact of vertical mixing on the directed differentiation of iPSCs into brain organoids and should be included in the revised manuscript. The author's data suggests that early cues are imparted into iPSCs that are retained long term during their differentiation into brain organoids.

Response to Reviewer #2 - comment 1)

>We greatly appreciate reviewer #2's constructive comments. According to Reviewer #2's comments, we added the description as below.

Discussion Page 8, Line 204 in highlighted version of the manuscript;

Furthermore, the early cues may be imparted into iPSCs in the vertical mixing culture system that are retained long-term during their differentiation into brain organoids.

2)Additionally, I have concerns relevant to the sc-RNA seq data as presented in Figure 5. As shown in panel A, there are 15 individual clusters, but it is unclear what cell types these clusters represent.

Response to Reviewer #2- comment 2)

>We greatly appreciate reviewer #2's constructive comments. According to this comment by Reviewer #2, we added the explanation in Methods as below. Recent studies have proven that

scRNA-seq enables the definition of cell types based on transcriptional traits of the cells. Thus, the cells belonging to the same cell cluster indicate the same or similar cell types.

Results, page 6, Line 146 in highlighted version of the manuscript;

UMAP is the conventional dimensionality reduction of the data matrix of gene expressions in each cell. In the above process, cell clusters were defined using K-means clustering on principal component analysis (PCA) space, and the number of clusters was decided using the elbow method.

3) It appears the author's data was analyzed with a high level of cellular resolution, which by itself is not necessarily incorrect, but it becomes more important to then specifically annotate each cell cluster. Based on panel B, the author's data suggests that there is a loss of deep cortical layer neurons in vertically mixed organoids as shown by loss of TBR1 and SOX5. Could the authors please comment on this result?

Response to Reviewer #2- comment 3)

>We greatly appreciate reviewer#2's constructive comments. According to Reviewer #2's comment, we added the description in the Discussion as below.

Discussion, Page10, line 239 in highlighted version of the manuscript;

In addition to the increase of GABAergic neurons, the results of single cell analysis provided us with some insight regarding the fact that vertical mixing altered the composition of the cell population with cortical markers. There was a decrease in the number of deep cortical layer neurons in vertically mixed organoids as shown by the loss of TBR1 and SOX5. The relative increase in the number of GABAergic neurons resulted in a relative decrease in TBR1-positive cells, which are highly expressed in excitatory neurons³⁸, and SOX5-positive cells, which are specifically expressed in corticofugal neurons³⁹. On the other hand, the number of BCL11B-positive cells was retained, as they represent certain subtypes of inhibitory neurons⁴⁰. These alterations have adopted a ventral identity by vertical mixing.

Ref 38. Sequerra, E. B., Miyakoshi, L. M., Fróes, M. M., Menezes, J. R. & Hedin-Pereira, C. Generation of glutamatergic neurons from postnatal and adult subventricular zone with pyramidal-like morphology. *Cereb Cortex* **20**, 2583-2591, doi:10.1093/cercor/bhq006 (2010).

Ref39. Lai, T. *et al.* SOX5 controls the sequential generation of distinct corticofugal neuron subtypes. *Neuron* **57**, 232-247, doi:10.1016/j.neuron.2007.12.023 (2008).

Ref 40. Nikouei, K., Muñoz-Manchado, A. B. & Hjerling-Leffler, J. BCL11B/CTIP2 is highly expressed in GABAergic interneurons of the mouse somatosensory cortex. *J Chem Neuroanat* **71**, 1-5, doi:10.1016/j.jchemneu.2015.12.004 (2016).

4) Another interpretation of their sc-data is that organoids generated vertical mixing alters the timing of neurodevelopment. Since GABAergic neurons develop at a later time point than excitatory neurons, it is possible that vertical mixing accelerates the timing of neurodevelopment in organoids. Could the authors provide more immunohistochemical analyses of vertical mixed organoids at earlier time points?

Response to Reviewer #2- comment 4)

>We greatly appreciate reviewer #2's constructive comments. We conducted additional experiments and added data that may suggest that vertical mixing did not speed up the differentiation, but rather specifically facilitated the promotion of GABAergic neuronal differentiation, as below.

Results, Page 7, Line 174 in highlighted version of the manuscript;

Furthermore, we evaluated the generation of excitatory neurons and GABAergic neurons along the time axis in orbital mixing and vertical mixing, and found that vertical mixing might facilitate the promotion of GABAergic neuronal differentiation (Fig. S5).

Supplementary Figure 5. Generation of GABAergic neurons and glutamatergic neurons in orbital mixing and vertical mixing.

GABAergic neurons stained with anti-GABA antibody were observed predominantly in organoids by vertical mixing (a), while glutamatergic neurons stained with anti-vGLT1 antibody were observed mostly in organoids by orbital mixing (b). scale bars = 50 μ m.

5) Lastly, the authors suggest their novel methodology to generate organoids via vertical mixing could be used to model neurological disorders. Although I agree that the inverted morphological phenomenon induced in organoids is highly interesting and can be used to understand basic principles of human brain development, I am skeptical that this model

could be applied to modeling neurological disorders since the inverted morphology phenotype is not physiological.

Response to Reviewer #2- comment 5)

>We greatly appreciate reviewer #2's constructive comments. We agree with Reviewer #2's comment that the inverted morphology phenotype is not physiological. However, after additional experiments, we found that the inverted organoids presented hippocampal neuron-markers, including PROX1, suggesting that the inverted organoids can be used to analyze neurological diseases with affected hippocampal neurons besides GABAergic neurons. We added the data in Figure 8 and Supplementary Figure 4b as below.

Figure 8

Figure 8. Analysis of Alzheimer's disease by using inverted brain organoids.

Images of Aβ oligomers in brain organoid of healthy control (a) and Alzheimer's disease with APP E693Δ mutation (b). Accumulation of Aβ oligomers was seen in Alzheimer's disease organoids.

Scale bars = 50 μm.

Supplementary Figure 4

Supplementary Figure 4. Gene expression by single-cell RNA sequencing

Single-cell RNaseq analysis demonstrated that NKX2.1-positive cells (a) and PROX1-positive

cells (b) were detected predominantly in organoids generated by vertical mixing.

REVIEWERS' COMMENTS:

Reviewer #2 (Remarks to the Author):

I have reviewed the second revised manuscript by Suong et al and believe the authors have not adequately addressed the first major comment that I raised in my last review, and therefore I cannot recommend this manuscript in its current form to be published in Communications Biology. As I recommended in my first review, the quantification data provided by the authors in the first review showing no change in the percentage of SOX2 positive cells when organoids were moved from vertical mixing to orbital mixing on day 15 was not included in the revised manuscript. This data is important as it suggests that the effects of vertical mixing imparts its main effects in inducing inverted morphology in organoids at some timepoint between Day -7 and Day 15. Although the interpretation of this data was included as a sentence in the discussion, it is unclear what meaning this sentence has since no data was provided in the revised manuscript. In addition, I recommended an experiment to initiate both vertical and orbital shaking at day 15 since this would be a more direct way to compare both methodologies, but this new data was not included in the revised manuscript. All other comments are adequately addressed.